# Inference-Time Decomposition of Activations (ITDA): A Scalable Approach to Interpreting Large Language Models

**Patrick Leask** [1]   **Neel Nanda**   **Noura Al Moubayed** [1]

## Abstract

Sparse Autoencoders (SAEs) are a popular method for decomposing Large Language Model (LLM) activations into interpretable latents, however they have a substantial training cost and SAEs learned on different models are not directly comparable. Motivated by relative representation similarity measures, we introduce Inference-Time Decomposition of Activation models (ITDAs). ITDAs are constructed by greedily sampling activations into a dictionary based on an error threshold on their matching pursuit reconstruction. ITDAs can be trained in 1% of the time of SAEs, allowing us to cheaply train them on Llama-3.1 70B and 405B. ITDA dictionaries also enable cross-model comparisons, and outperform existing methods like CKA, SVCCA, and a relative representation method on a benchmark of representation similarity. Code available at github.com/pleask/itda.

## 1. Introduction

Mechanistic interpretability aims to reverse-engineer neural networks into human-interpretable algorithms (Olah et al., 2020; Meng et al., 2022; Geva et al., 2023; Nanda et al., 2023; Elhage et al., 2021). Sparse autoencoders (SAEs) have recently emerged as a promising alternative for decomposing LLM activations into a dictionary of interpretable and monosemantic latents (Cunningham et al., 2023; Bricken et al., 2023; Gao et al., 2025; Marks et al., 2025; Lieberum et al., 2024; Rajamanoharan et al., 2024). However, training SAEs is computationally expensive: it requires model activations for hundreds of millions or even billions of tokens, which are expensive to collect for large models; and the parameter count of SAEs can exceed that of the models to which they are applied (Sharkey et al., 2025). As a result, much academic research relies on open-

source SAEs, such as Lieberum et al. (2024) and He et al. (2024), which are currently limited to models with up to 27B parameters.

Furthermore, comparing SAEs trained on different models is challenging: their parameters are learned from model activations, which vary for the same input between models even of the same family. Lan et al. (2024) proposes that SAEs reveal universal feature spaces across LLMs by using centered kernel alignment (Kornblith et al., 2019) to compare the decoders of SAEs. The problem of aligning the representation spaces of different LLMs is also fundamental to absolute measures of representation similarity such as SVCCA (Raghu et al., 2017) and CKA (Kornblith et al., 2019), which is addressed with relative representation measures by Moschella et al. (2022), who use the cosine similarity of representations with respect to the representations of a fixed random set of anchor inputs in different models.

In this paper, we introduce Inference-Time Decomposition of Activations (ITDA) as a lightweight alternative to SAEs. Similarly to the relative representation method of Moschella et al. (2022), we construct a dictionary of anchors to which we relate activations by their cosine similarity. However, we extend this method by decomposing activations into this dictionary using matching pursuit, a method for inference-time optimization. As with SAEs, this approach finds sparse decompositions of test LLM activations. In contrast to Moschella et al. (2022), who use random anchor points, we form the dictionary by initializing with the most frequent activations, and greedily adding training activations that are most poorly reconstructed by the dictionary so far.

ITDAs can be trained on a million tokens to the same performance as SAEs trained on hundreds of millions of tokens, resulting in a proportional decrease in training times. GPT-2 (Radford et al., 2019) SAEs take hours to train (Leask et al., 2025), whereas ITDAs can be trained in minutes to similar reconstruction performance. This means we are able to train ITDAs on 70B and 405B parameter LLMs: an order of magnitude greater than the largest models on which open-source SAEs have been trained. The cross entropy loss degradation when replacing activations with their ITDA reconstructions is similar or slightly worse to using SAE reconstructions

[1]Department of Computer Science, Durham University. Correspondence to: Patrick Leask <patrickaaleask@gmail.com>.

*Proceedings of the $42^{nd}$ International Conference on Machine Learning*, Vancouver, Canada. PMLR 267, 2025. Copyright 2025 by the author(s).

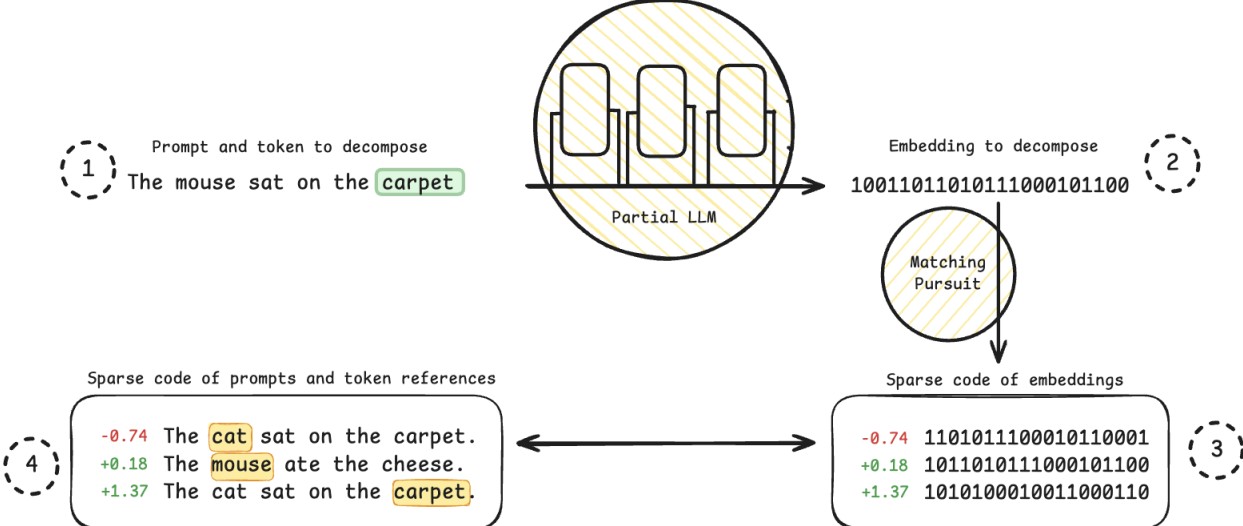

Figure 1. Stylized system-level diagram of inference-time decomposition of activations. To decompose a target token in a prompt (1), we compute its activation at a certain point in the LLM (2), for example after layer 8 in GPT-2. We use matching pursuit to decompose the activation into a sparse code of other activations collected from the same point in the LLM (3). Our dictionary atoms are labeled with the prompt and token for which the activation was collected, which allows direct interpretation of the sparse code (4).

on Pythia models (Biderman et al., 2023), however substantially worse on Gemma-2 (Team et al., 2024).

SAE latents are learned by gradient descent, meaning that latents learned on two different models have no inherent relation. On the other hand, an ITDA latent is the activation of a model at a certain point for a specific prompt and token. So, although the activations of the two models cannot be directly compared, we can compare the prompt-tokens pairs constituting each dictionary. We use this to compare the representation spaces of different models. In particular, we introduce an ITDA dictionary difference measure for representation similarity based on the Jaccard similarity, or intersection over union (IoU), and demonstrate that it achieves state-of-the-art performance on a layer-matching task due to Kornblith et al. (2019), as well as reproducing results on layer freezing from Raghu et al. (2017). Since ITDA dictionaries are useful for interpretability and their differences reflect changes in model representation spaces, we see exciting potential for their use in identifying behavioral differences between models, as proposed in (Lindsey et al., 2024). For example, finding concerning changes in model behavior when performing chat fine-tuning, such as increases in scheming or sycophancy. However, we do not explore model diffing in this paper, leaving it for future work.

In summary, we introduce Inference-Time Decomposition of Activations (ITDAs) as an alternative to SAEs, offering significant advantages in efficiency and scalability:

- ITDAs are 100x faster to train than SAEs, and only require around 1% of the training tokens. As a result, we trained them on 70B and 405B LLMs, an order of magnitude larger than any LLMs on which open source SAEs have been trained.

- They achieve similar reconstruction performance to SAEs on some models, though worse on others, and have comparable automated interpretability scores.

- ITDA dictionaries transfer readily between LLMs in a way that SAEs do not; and we use this to construct a representation similarity metric that is state-of-the-art on a benchmark for similarity indices.

- ITDA atoms have interpretable labels, i.e. the prompt and token from which the activation was taken. This gives their atoms some inherent interpretability, unlike SAE latents.

## 2. Related Work

In this section, we discuss the recent advancements in applying sparse dictionary learning methods to interpret the representations of LLMs. We focus on SAEs, rather than other sparse dictionary learning methods, as they are the primary subject of the mechanistic interpretability literature (Bricken et al., 2023; Cunningham et al., 2023; Dunefsky

et al., 2024; Gao et al., 2025; Lan et al., 2024; Lieberum et al., 2024; Lindsey et al., 2024; Makelov et al., 2025; Marks et al., 2025; Leask et al., 2025). We provide a glossary of terms in Appendix A.1.

## 2.1. Sparse Autoencoders (SAEs)

Sparse dictionary learning is the problem of finding a decomposition of a signal that is both sparse and overcomplete (Olshausen & Field, 1997). Lee et al. (2007) applied the sparsity constraint to deep belief networks, with SAEs later being applied to the reconstruction of neural network representations (Bricken et al., 2023; Cunningham et al., 2023). In the context of LLMs, SAEs decompose model representations $\mathbf{x} \in \mathbb{R}^n$ into sparse linear combinations of learned latents. It is hoped that these latents correspond to monosemantic and interpretable features of the representations, and that the sparse codes for a specific representation are also interpretable.

An SAE consists of an encoder and a decoder:

$$\mathbf{f}(\mathbf{x}) := \sigma(\mathbf{W}^{\text{enc}}\mathbf{x} + \mathbf{b}^{\text{enc}}), \qquad (1)$$

$$\hat{\mathbf{x}}(\mathbf{f}) := \mathbf{W}^{\text{dec}}\mathbf{f} + \mathbf{b}^{\text{dec}}. \qquad (2)$$

where $\mathbf{f}(\mathbf{x}) \in \mathbb{R}^m$ is the sparse latent activations and $\hat{\mathbf{x}}(\mathbf{f}) \in \mathbb{R}^n$ is the reconstructed input. $\mathbf{W}^{\text{enc}}$ is the encoder matrix with dimension $n \times m$ and $\mathbf{b}^{\text{enc}}$ is a vector of dimension $m$; conversely $\mathbf{W}^{\text{dec}}$ is the decoder matrix with dimension $m \times n$ and $\mathbf{b}^{\text{dec}}$ is of dimension $n$. The activation function $\sigma$ enforces non-negativity and sparsity in $\mathbf{f}(\mathbf{x})$, and a latent $i$ is active on a sample $\mathbf{x}$ if $f_i(\mathbf{x}) > 0$.

SAEs are trained on the representations of a language model at a particular site, such as the residual stream, on a large text corpus, using a loss function of the form

$$\mathcal{L}(\mathbf{x}) := \underbrace{\|\mathbf{x} - \hat{\mathbf{x}}(\mathbf{f}(\mathbf{x}))\|_2^2}_{\mathcal{L}_{\text{reconstruct}}} + \underbrace{\lambda \mathcal{S}(\mathbf{f}(\mathbf{x}))}_{\mathcal{L}_{\text{sparsity}}} + \alpha \mathcal{L}_{\text{aux}} \qquad (3)$$

where $\mathcal{S}$ is a function of the latent coefficients that penalizes non-sparse decompositions, and $\lambda$ is a sparsity coefficient, where higher values of $\lambda$ encourage sparsity at the cost of higher reconstruction error. Some architectures also require the use of an auxiliary loss $\mathcal{L}_{\text{aux}}$, for example to recycle inactive latents in TopK SAEs (Gao et al., 2025). We expand on the different SAE variants in Appendix A.2. We also note that Nanda et al. (2024) applied inference-time optimization to a pre-trained SAE dictionary, though their approach is otherwise very different to ours.

While SAEs have been a central approach to learning sparse features in neural networks, there is a long history of sparse

dictionary learning and inference-time approaches outside of autoencoders. Classical methods such as K-SVD (Aharon et al., 2006) and Matching Pursuit (Mallat & Zhang, 1993) iteratively choose dictionary elements to represent a signal under an $\ell_0$ or $\ell_1$ constraint. These approaches typically proceed by directly solving a sparse coding objective (e.g., via Orthogonal Matching Pursuit or Iterative Shrinkage-Thresholding methods like FISTA (Beck & Teboulle, 2009)) instead of learning an explicit encoder network. In many computer vision tasks, these traditional dictionary-learning techniques have shown good performance on denoising and inpainting (Elad & Aharon, 2006), but can be computationally expensive at inference time. More recent methods have also investigated structured sparsity (e.g. group sparsity) or used deep dictionary learning frameworks that unroll sparse coding iterations into learned networks (Gregor & LeCun, 2010). The method we propose in Section 3 has more in common with these approaches than the SAEs used so far in mechanistic interpretability.

## 2.2. SAEs for Mechanistic Interpretability

SAEs have been demonstrated to recover sparse, monosemantic, and interpretable features from language model representations (Bricken et al., 2023; Cunningham et al., 2023; Templeton, 2024; Gao et al., 2025; Rajamanoharan et al., 2025; 2024), however their application to mechanistic interpretability is nascent. After training, researchers often interpret the meaning of SAE latents by examining the dataset examples on which they are active, either through manual inspection using features dashboards (Bricken et al., 2023) or automated interpretability techniques (Gao et al., 2025). SAEs have been used for circuit analysis (Marks et al., 2025) in the vein of (Olah et al., 2020; Olsson et al., 2022); to study the role of attention heads in GPT-2 (Kissane et al., 2024a); and to replicate the identification of a circuit for indirect object identification in GPT-2 (Makelov et al., 2025). Transcoders, a variant of SAEs, have been used to simplify circuit analysis and applied to the greater-than circuit in GPT-2 (Dunefsky et al., 2024). Whilst the application of SAEs to mechanistic interpretability is supported by qualitative and quantitative evidence, their usefulness is highly dependent on hyperparameterisation (Leask et al., 2025). (Nanda et al., 2024) propose the use of inference-time optimization on sparse autoencoder decoder matrices to quickly evaluate and compare them.

Sharkey et al. (2025) list the computational cost of training SAEs as an open problem in their application to decomposing model representations. In particular, they note that the number of parameters in an SAE can exceed the number of parameters in the LLM whose representations they decompose. For example, in Lieberum et al. (2024), the largest of the SAEs trained on Gemma 2 2B have almost 5 billion parameters, compared to only 2 billion parameters in the

LLM itself.

## 2.3. Comparing Models

A range of methods for comparing representations between neural networks has been developed. Inspired by Erhan et al. (2010), Olah (2015) applied t-SNE, a dimensionality reduction technique (Van der Maaten & Hinton, 2008), to the representations of vision and language models. Lenc & Vedaldi (2015); Bansal et al. (2021) stitched layers of two frozen models with a trained intermediate adapter layer, and evaluated the similarity of the model's representations by the performance of the stitched model. Representation similarity metrics compare the alignment of the representation subspaces of different models, and include Singular Vector Canonical Correlation Analysis (SVCCA) (Raghu et al., 2017) and Centered Kernel Alignment (CKA) (Kornblith et al., 2019). The performance of linear probes trained on the representations of different models can provide insight into what information the representations represent (Alain, 2016; Hewitt & Manning, 2019). Li et al. (2016) investigated whether different neural networks converge to the same representations, and (Garipov et al., 2018; Zhao et al., 2020) find that different models are occupied by low-loss paths in the parameter space. Olah et al. (2020) provide examples of potential universal features, such as curve detectors, in vision models, and Olsson et al. (2022) find evidence of induction heads in language models of different sizes. Bricken et al. (2023) found similar SAE latents in different models, and Kissane et al. (2024b) found examples of SAEs that transfer between base and fine-tuned versions of the same language model. Lindsey et al. (2024) used Crosscoders, SAEs trained on the representations of multiple models, to find features present in a fine-tuned version of an LLM that were not present in the base model. Relative representation methods are kernel methods (Hofmann et al., 2008) that measure similarity against a set of prototype inputs (Moschella et al., 2022), which avoids learning model specific parameters from absolute model representations.

# 3. Inference-Time Decomposition of Activations (ITDA)

We introduce a novel method for decomposing model activations into interpretable units, which we call Inference-Time Decomposition of Activations (ITDA).

## 3.1. Absolute and Relative Representations

Given a training dataset $\mathbf{x} \in \mathbf{X}$, LLMs learn an embedding function $E_\theta : \mathbf{X} \to \mathbb{R}^d$, where $\theta$ is the parameters of the model. $E_\theta$ maps each sample $\mathbf{x}^{(i)} \in \mathbf{X}$ to its absolute latent representation $e^{(i)} = E_\theta(\mathbf{x}^{(i)})$. This absolute representations are learned by optimization over an objective function:

$$\min_\theta \mathbb{E}_{\mathbf{x} \sim \mathbf{X}}[L(E_\theta(\mathbf{x})) + \text{Reg}(\theta)]$$

Where Reg is a regularization function on the parameters $\theta$. These representations are different between models: two models that are behaviorally identical may have representations that are rotations or affine transformations $T$ of the other (Kornblith et al., 2019; Raghu et al., 2017). SAEs are trained on these representations so, if their decoders do describe feature spaces in LLMs (Lan et al., 2024), then they are also subject to the same symmetries as absolute representations.

Moschella et al. (2022) propose that, whilst the absolute representations change between models, the angles between elements of the representation space remain the same. That is, for a transformation $T$, $\angle(\mathbf{e}^{(i)}, \mathbf{e}^{(j)}) = \angle(T\mathbf{e}^{(i)}, T\mathbf{e}^{(j)})$ for every $\mathbf{x}^{(i)}, \mathbf{x}^{(j)} \in \mathbb{X}$. Based on this assumption, they construct their representation by selecting a subset $\mathcal{A}$ of anchor points of the training data $X$. Each sample in the training data is represented with respect to the embedded anchors $\mathbf{e}^{(j)} = E(\mathbf{a}^{(j)})$ with $\mathbf{a}^{(j)} \in \mathcal{A}$, using a similarity function $\text{sim} : \mathbb{R}^d \times \mathbb{R}^d \to \mathbb{R}$. Cosine similarity $S_C$ is used as this similarity function as it preserves angles:

$$S_C(\mathbf{a}, \mathbf{b}) = \frac{\mathbf{a} \cdot \mathbf{b}}{||\mathbf{a}||||\mathbf{b}||} \qquad (4)$$

## 3.2. Inference Time Sparse Coding

We replace the learned linear encoder of SAEs with a dictionary learning approach to solve the sparse coding problem. For inputs $\mathbf{x} \in \mathbb{R}^d$, we maintain a dictionary of $n$ activations $\mathbf{D} \in \mathbb{R}^{n \times d}$, which may not necessarily be basis vectors in $\mathbb{R}^d$. We then solve the following sparse coding problem to obtain the coefficients $\mathbf{a}$:

$$\min_{\mathbf{a} \in \mathbb{R}^n} ||\mathbf{x} - \mathbf{aD}|| \text{ subject to } ||\mathbf{a}||_0 \leq L_0 \qquad (5)$$

where $||\cdot||$ is the $l_0$-pseudo-norm (the number of non-zero elements), and $L_0$ is a pre-specified sparsity level (the number of latents used to represent each $\mathbf{x}$).

We obtain approximate sparse codes $\mathbf{a}$ with an Matching Pursuit (MP) solver (Mallat & Zhang, 1993). For an input vector $\mathbf{x}$ we construct a solution by the following algorithm, which is fully described in Appendix Algorithm 2.

1. Selecting, at each iteration, the dictionary activation $\mathbf{d}_j$ whose correlation with the residual is the largest in magnitude.

2. Updating the residual by subtracting the projection of the residual onto the newly chosen activation.

3. Repeating for $L_0$ steps to achieve the desired sparsity.

Thus, the encoder step of the ITDA can be written as $\mathbf{a} = \text{MP}(\mathbf{x}, \mathbf{D}, L_0)$ and the decoder as $\hat{\mathbf{x}} = \mathbf{aD}$. Matching Pursuit uses correlation when choosing the next dictionary activation to include in the sparse code, which is equal to the unnormalised cosine similarity between the activations.

### 3.3. Dictionary learning

Moschella et al. (2022) randomly sample their anchors from each class in the training set. However, comparison of randomly sampled dictionaries, as is done with SAE decoders in (Lan et al., 2024). Therefore we deterministically and iteratively construct the dictionary by attempting to reconstruct activations from the existing dictionary, and adding them to the dictionary if the reconstruction loss is above a threshold. This contrasts with SAEs, which have a fixed sized dictionary, with the choice of dictionary size largely determining the reconstruction performance. In ITDA, we choose the loss threshold, which determines the size of the dictionary.

Concretely, for a new sample $\mathbf{x}$, its reconstruction $\hat{\mathbf{x}}$, and the current dictionary $\mathbf{D}$, we define the reconstruction loss $l(x)$ as the mean-squared error between the sample and its reconstruction:

$$l(\mathbf{x}) = ||\mathbf{x} - \hat{\mathbf{x}}||_2^2 \tag{6}$$

If $l(\mathbf{x})$ exceeds a chosen target loss threshold $\tau$, we add the normalized $\mathbf{x}$ to the dictionary. In practice, when we construct these dictionaries, we batch inputs. If there are identical or similar inputs in a batch, then it is possible that they are both added. As such, we additionally filter the dictionary for repeated activations after construction. The full algorithm is displayed in Algorithm 1. In general, a lower value of $\tau$ leads to a larger dictionary size and lower reconstruction loss than high values of $\tau$.

### 3.4. Interpretable Labels

We label the selected dictionary atoms with the prompt and token to which the activation corresponds. This is not used for decomposition, but provides an additional property of inherent interpretability. Gaining any understanding of SAE latent functionality requires automated interpretability, or other further experimentation, whilst with ITDA latents we already have these interpretable labels. We also use the labels, rather than the activations themselves, in our representation similarity study in Section 5.

---

**Algorithm 1** ITDA Training with Inference-Time Optimization (ITO)

---

**Input:** Training data $\{x_i\}$, initial dictionary $\mathbf{D}$ (optional), sparsity level $L_0$, threshold $\tau$
**Output:** Learned dictionary $\mathbf{D}$
Initialize dictionary $\mathbf{D}$ (e.g., by selecting common activations or random sampling)
**for** each batch $\mathcal{B} = \{x_1, x_2, \ldots, x_B\}$ from training data **do**
    **for** each sample $x$ in $\mathcal{B}$ **do**
        Compute sparse code $\mathbf{a} = \text{OMP}(x, \mathbf{D}, L_0)$
        Reconstruct $\hat{\mathbf{x}} = \mathbf{aD}$
        Compute reconstruction loss $\ell(x) = ||x - \hat{\mathbf{x}}||_2^2$
        **if** $\ell(x) > \tau$ **then**
            Add $x$ to dictionary: $\mathbf{D} \leftarrow \mathbf{D} \cup \{x\}$
        **end if**
    **end for**
    Remove duplicate atoms in $\mathbf{D}$ (optional)
    Normalize dictionary: $\mathbf{d}_j \leftarrow \mathbf{d}_j / ||\mathbf{d}_j||_2$ for all rows $j$
**end for**
Return learned dictionary $\mathbf{D}$

---

## 4. Comparison with SAEs

We first evaluate on the type of sparse coding problem for which SAEs are used: namely, the decomposition and reconstruction of LLM activations.

### 4.1. Reconstruction Performance

We trained ITDAs on the residual stream activations of two Pythia models (Biderman et al., 2023) and the two billion parameter variant of Gemma 2 (Team et al., 2024) on a subset of the Pile dataset (Gao et al., 2020) limited to 128 tokens. We evaluate these in terms of their cross entropy loss score (CE loss score):

$$\frac{H^* - H_0}{H_{\text{orig}} - H_0} \tag{7}$$

where $H_{\text{orig}}$ is the original cross entropy loss of the LLM, $H^*$ is the cross entropy loss of the LLM when its activations are replaced with the SAE reconstruction of the activation, and $H_0$ is the cross entropy score when zero-ablating the model activations. The cross entropy is calculated on a text-prediction on the Pile dataset (Gao et al., 2020).We use SAEBench (Karvonen et al., 2024a), an SAE benchmarking suite, for these evaluations. Direct comparison between SAE types is challenging as most existing types do not have fixed $L_0$s, and ITDAs do not have fixed dictionary sizes (See Table 2). However, for purpose of comparison we crop the ITDA dictionary to a fixed size.

Figure 7(a) shows the performance of ITDAs in comparison

to SAEs. On Pythia models, ITDAs generally perform better than ReLU SAEs, the original SAEs used in (Bricken et al., 2023; Cunningham et al., 2023), but worse than the more recent top-k SAEs (Gao et al., 2025). On Gemma 2 2B, ITDAs perform substantially worse than SAEs, where pretrained SAEs are available.

However, the SAEs were trained on hundreds of millions of tokens. In comparison, the ITDAs were trained to their maximum performance on 1.2 million tokens. Where compute and training dataset are not a constraint, SAEs can achieve better performance than ITDAs; but for most applications this is not the case. Sharkey et al. (2025) cites the high cost of training SAEs as a major open problem in decomposition of activations using sparse dictionary learning. SAE performance increases with the size of the dictionary, and one of the major factors in choosing dictionary size is the training cost: (Leask et al., 2025) trained SAEs on GPT-2 small ranging from 768 latents to 98304 latents, which took between 2 and 8 hours to train. Given the far lower cost of training ITDAs, it is possible to train much larger ITDAs than SAEs to achieve similar reconstruction performance. However, this choice of dictionary size impacts interpretability in both SAEs (Leask et al., 2025; Karvonen et al., 2024b) and ITDAs.

We include interpretability benchmark results from SAEBench in Appendix A.3.4. However, we emphasize caution in interpreting these results due to the novelty of automated benchmarking for SAEs, and their questionable applicability to methods other than SAEs; concerns that we expand on in the appendix.

### 4.2. Llama Case Study

We trained and release ITDAs on the 70 billion and 405 billion parameter versions of Llama 3.1 (Dubey et al., 2024) at [redacted]. These ITDAs were trained on a consumer GPU using a dataset of 1 million activations collected with (Fiotto-Kaufman et al., 2024). Existing open-source SAEs are limited to a maximum LLM parameter count of 27 billion such as in Lieberum et al. (2024). Evaluating these large models using benchmarks like SAEBench (Karvonen et al., 2024a) is currently infeasible due to the high computational cost of collecting activations, as opposed to training or running ITDAs. We provide these case-studies to demonstrate that ITDAs find interpretable and monosemantic features in the activations of these large models.

We trained an ITDA with 17939 latents on layer 40 of Llama 3.1 70B, and collected the coefficients (which we call activations) of each latent (which we call latents) on activations of the LLM on a 10,000 prompt sample of the Pile dataset (Gao et al., 2020). Similarly to SAEs (Bricken et al., 2023), we interpret the feature to which a latent corresponds by considering the prompts on which it most strongly activates.

We cherry-picked three examples of latents that activate in interestingly different cases, and interpret the feature they represent based on the prompts and tokens on which they strongly activate.

1. Latent 50 responds specifically to the token "How"

2. Latent 16990 responds to tokens relating to surprise and amazement.

3. Latent 17002 responds to context relating to offers of help, particularly helping students with homework.

Activation histograms and top-activating inputs are provided in Appendix A.3.1.

We assigned interpretable labels - the prompt and token to which an activation relates - to the atoms. For example, latent 16990 corresponds to the token "surprising", and the top activating latents correspond to prompt tokens relating to "surprise". Direct interpretation of the atom labels can be misleading though, as is the case for latent 17002. Latent 17002 corresponds to the presence of "a" in its prompt, but its top activating inputs relate to "helping students with their homework". Whilst this is also the context of the prompt corresponding to Latent 17002, it is not immediately obvious whether the token or the context of the token is relevant.

Another key difference with ITDAs is the existence of negative activations. Whilst large positive activations of ITDA latents correspond to monosemantic and interpretable features of the data, small positive and negative examples are difficult to interpret, however this is also the case with SAEs for small positive values (Bricken et al., 2023). Large negative activations are exceptionally rare, constituting only 0.000743% of activations on our dataset.

Appendix A.3.2 contains a number of examples of prompt decompositions using this ITDA.

## 5. Representation Similarity

Identical LLMs trained on the same data but with different initialisations can learn representations that are rotated or at different scales. Measuring the similarity of their representations generally requires learning linear maps to handle these transformations (Kornblith et al., 2019; Raghu et al., 2017). Moschella et al. (2022) propose a relative representation measure, where representations are assigned to their nearest neighbor in a set of anchor points (detailed in Section 3.1), and therefore does not require learning a map between the representation spaces. These approaches compare datasets of representations, which can be large to provide a meaningful sample.

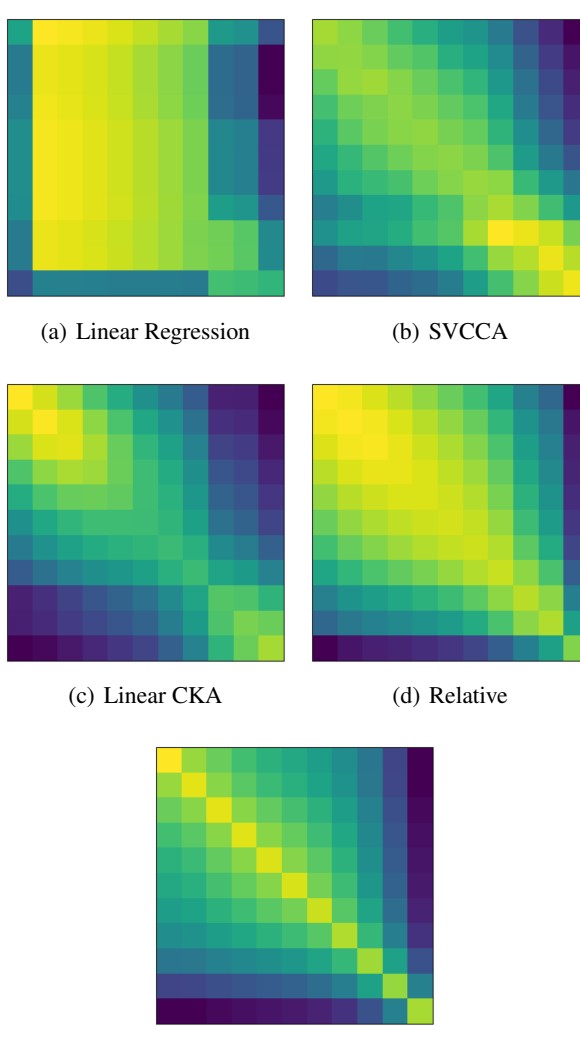

(a) Linear Regression      (b) SVCCA

(c) Linear CKA      (d) Relative

(e) ITDA

*Figure 2.* Heatmap of similarity indexes between layers in instances of GPT-2 small with random initialisations, metric average across all pairs of model. Each axis represents the ordered layers of the model. The dark blue colors represent pairs of layers that, on average, have low similarity scores, whilst the yellow colors represent pairs of layers that, on average, have high similarity scores. Note that these are averages, and the scores for specific comparisons vary, but in general, higher contrast between values in the same row and column indicate that the metric is better able to differentiate between layers. See Figure A.4 for GPT-2 medium results.

SAE dictionaries have been proposed as universal descriptions of representation space (Lan et al., 2024). The similarity of two representation spaces can be measured by learning a map between the dictionaries of two SAEs. Where SAEs are available, this provides an alternative to computing similarity on representations. However, SAEs themselves are expensive to train, and the learning of this map is similar to other absolute representation metrics.

We propose the Jaccard Index (Intersection over Union) of two ITDA dictionaries as a measure of representation similarity. Concretely, for models $M_0$ and $M_1$, we construct dictionaries $D_0$ and $D_1$ as described in Algorithm 1. Then our measure for representation similarity is:

$$S(M_0, M_1) := \frac{|D_0 \cap D_1|}{|D_0 \cup D_1|} \qquad (8)$$

This approach is a relative representation measure, so it is not necessary to learn a map between the representation spaces. ITDAs are cheap to train, and the Jaccard Similarity is trivial to compute, and so this approach further resolves the challenge with using SAEs as descriptions of feature spaces.

We evaluate this method on two tasks: first, we match layers in different instances of the same model by their maximum similarity as in Kornblith et al. (2019); then we perform this task at the model level, attempting to match different instances of the same model using the union of their ITDA dictionaries. We also reproduce results in layer convergence during training in Appendix A.4.2.

### 5.1. Model Instance Layer Similarity

Kornblith et al. (2019) introduces a benchmark for similarity indices based on a layer matching task. For two architecturally identical models trained from different initialisations, we expect that the value of the residual stream after layer $i$ in the first model is most similar to the value of the residual stream after layer $i$ in the second model, rather than after some other layer $j \neq i$. Similarity indices are scored on their accuracy on this task. That is, the score of a similarity index $S$ is defined as:

$$\frac{1}{M^2(n+1)} \sum_{m,m' \in \mathcal{M}} \sum_{i=0}^{n} \mathbf{1}\left[\arg\max_{j} S(m_i, m'_j) = i\right] \qquad (9)$$

Where $\mathcal{M}$ is the set of models of size $M$ and $n$ is the number of layers in each model with $m_i, 0 \le i < n$ is the partial model after a certain layer, and $\mathbf{1}[\cdot]$ is the indicator function.

We replicate this experiment on two sets of five instances of the GPT-2 small and GPT-2 medium taken from the Mistral

package (Karamcheti et al., 2021).

Table 2 shows the performance of CKA (Kornblith et al., 2019), SVCCA (Raghu et al., 2017), and ITDA on this task. Figure 5 shows the layer similarity between all pairs of layers, averaged across all pairs of models, for the GPT-2 small. Figure A.4 shows the same information for the GPT-2 medium instances.

We emphasize that ITDAs offer significantly reduced training costs, and the use of the simple Jaccard similarity index makes this analysis much more efficient compared to using SAEs. Training each ITDA on GPT-2 small took an average of 8 minutes, in comparison to two to eight hours for each SAE (depending on dictionary size, on comparable hardware).

### 5.2. Model Similarity

The ITDAs used to calculate layer similarity are trained at specific layers within the model, which resulted in their dictionaries forming universal descriptions of the representation space at that layer. We briefly test whether the union of ITDA dictionaries over all layers within a model forms a universal description of the representation space for the entire model. That is, for an ITDA dictionary $D_{0,i}$ at layer $i$ of model 0, we say

$$D_0 = \bigcup_{i=0}^{L} D_{0,i} \qquad (10)$$

where L is the layers of model 0. We test whether we can distinguish between GPT-2 small and GPT-2 medium using the Jaccard similarity of dictionaries.

The Jaccard similarity between different instances of the same model architecture (either GPT-2 small or GPT-2 medium) ranges from 0.56 to 0.59; whereas the Jaccard similarity between different model architectures range from 0.46 to 0.47 (Figure 11. This suggests the dictionaries do track the difference between the models, and suggests further research into model differencing using ITDAs may be interesting.

Using CKA or SVCCA to perform a global layer matching task involves concatenating the activation vectors at all residual stream locations. For GPT-2 Medium this is a vector with 24,576 elements per token introducing substantial training cost.

## 6. Discussion

We introduced Inference-Time Decomposition of Activations as a scalable approach to LLM interpretability. ITDAs can perform as well as SAEs on certain LLMs, whilst being a 100x faster to train. Due to the reduced training cost

of this method, we are able to train ITDAs on LLMs that are an order of magnitude larger than the largest LLMs on which open-source SAEs have been trained. We provided examples of ITDA latents for these large models, to demonstrate that ITDAs learn interpretable and monosemantic features similarly to SAEs. Unlike SAEs decoders, ITDA dictionaries transfer readily between LLMs because they are associated with a prompt and token index, rather than being learned from the representations of a specific model. We used this property to construct an index for representational similarity that is state-of-the-art on a LLM layer matching task.

The primary limitation of our approach is that ITDA performs worse on reconstruction and interpretability tasks than recent types of SAE, such as TopK and P-Annealing SAEs; however, they do often perform comparably to ReLU SAEs. Whilst it is possible that further refinement of the ITDA approach will lead to better reconstruction and interpretability results, we do not currently suggest ITDAs broadly replace SAEs on existing applications such as circuit discovery (Dunefsky et al., 2024; Marks et al., 2025), where SAEs are publicly available for those models. However, we are keen to see ITDAs applied more to tasks where training SAEs is prohibitively expensive such as very large models, or where large numbers of SAEs are required, such as in our layer freezing experiments or on checkpoints during model training. We are also excited to see further refinement of the ITDA learning algorithm: in particular, we think data ordering during training could be very important for auto-regressive LLMs.

Our results demonstrate that ITDAs find interpretable and monosemantic features in LLM representations, and that the difference in their dictionaries is an accurate measure of representation similarity. We believe that this makes ITDAs an ideal candidate for further research into finding differences between models, such as base and fine-tuned LLMs. Lindsey et al. (2024) suggest that model diffing is an important component of deploying iterative models, as is done by API model providers: training SAEs for all fine-tunes of a model is likely too costly, and ITDAs offer an efficient alternative.

## Impact Statement

This paper presents work whose goal is to advance the field of mechanistic interpretability, rather than machine learning in general. Whilst there are potential societal consequences to the advancement of the ML field, we feel that research that improves understanding of ML models is unlikely to additionally contribute to these consequences, rather giving the field more tools to avert them.

| Metric | GPT-2 Small | GPT-2 Medium |
|---|---|---|
| Linear Regression (baseline) | 0.16 | 0.07 |
| SVCCA (Raghu et al., 2017) | 0.50 | 0.44 |
| Linear CKA (Kornblith et al., 2019) | 0.69 | 0.61 |
| Relative (Moschella et al., 2022) | 0.87 | 0.78 |
| ITDA (ours) | **0.88** | **0.89** |

*Table 1.* Accuracy of Linear CKA, SVCCA, and ITDA on the layer matching task for GPT-2 small and medium. ITDA Jaccard similarity outperforms all other methods, including the other relative representation approach.

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

# A. Appendix

## A.1. Glossary

**Activation:** For part of a model $M$ and an input sequence $\mathbf{x}$, an activation is $\mathbf{a} = M(\mathbf{x})$. A partial model could be, for example, the residual stream value for an input after the 8th layer of GPT-2. We are interested in the activation for a specific token within the prompt, rather than the activation matrix for an entire sequence. **Activations** and **representations** are used interchangeably, as **activation** is the dominant term in mechanistic interpretability literature and **representation** is the dominant term in representation similarity literature.

**Atom:** A component of the dictionary learned by sparse dictionary learning. Examples are decomposed into sparse codes of these atoms. The **atom** terminology is more commonly used in classical sparse dictionary learning literature, whereas **latent** is more common in contemporary mechanistic interpretability SAE literature.

**Feature:** A true but unknown property of a data-point, contrasts with **Latent**.

**Latent:** Refers to the components of SAEs that are used in sparse codes to decompose activations. Ideally, these are the same as **features**, but empirically this is not necessarily the case. See **atom**.

**Latent activation:** The coefficient of a latent or atom in a sparse code.

**Representation:** See **activation**.

**Sparse code:** For an example $\mathbf{x}$ and an encoding function $\mathbf{f}$ for an SAE with dictionary size $m$, $\mathbf{f}(\mathbf{x}) \in \mathbb{R}^m$ is a sparse code. The sparsity, i.e. number of zero or non-zero terms, is determined by the optimization process and activation function.

**Token:** In the context of LLMs, a token is a discrete unit of text—such as a word, subword, or punctuation—that the model processes and generates during language understanding and generation.

## A.2. SAE Variants

| Method | Fixed $L_0$ | Fixed Dictionary Size |
|---|---|---|
| ReLU | ✗ | ✓ |
| TopK | ✓ | ✓ |
| JumpReLU | ✗ | ✓ |
| ITDA | ✓ | ✓/✗ |

*Table 2.* Comparison of SAE types with respect to fixed $L_0$ and fixed dictionary size. When constructed by the method described in Section 3, ITDAs have a variable dictionary size, however they can be cropped to a desired size.

**ReLU SAEs** (Bricken et al., 2023) use the L1-norm $S(\boldsymbol{f}) := ||\boldsymbol{f}||_1$ as an approximation to the L0-norm for the sparsity penalty. This provides a gradient for training unlike the L0-norm, but suppresses latent activations harming reconstruction performance (Rajamanoharan et al., 2025). Furthermore, the L1 penalty can be arbitrarily reduced through reparameterization by scaling the decoder parameters, which is resolved in Bricken et al. (2023) by constraining the decoder directions to the unit norm. Resolving this tension between activation sparsity and value is the motivation behind more recent architecture variants.

**Gated SAEs** (Rajamanoharan et al., 2025) separate the selection of dictionary elements used for reconstruction, with the estimation of the coefficients of these dictionary values. This results in the following architecture:

$$\pi_{\text{gate}}(\mathbf{x}) := W_{\text{gate}}(\mathbf{x} - \mathbf{b}_d) + \mathbf{b}_{\text{gate}} \tag{11}$$

$$\boldsymbol{f}(\boldsymbol{x}) := \mathbb{I}[\pi_{\text{gate}}(\boldsymbol{x}) > 0] \odot \text{ReLU}(W_{\text{mag}}(\boldsymbol{x} - \boldsymbol{b}_d) + \boldsymbol{b}_m ag) \tag{12}$$

$$\hat{x}(\boldsymbol{f}(\boldsymbol{x})) = W_d \boldsymbol{f}(\boldsymbol{x}) + \boldsymbol{b}_d \tag{13}$$

Where $\mathbb{I}[\cdot > 0]$ is the Heaviside step function and $\odot$ donates elementwise multiplication.

**P-Annealing SAEs** (Karvonen et al., 2024b) replace $L_1$ minimisation with $L_p$ minimisation, where $p < 1$ and is decreasing throughout training. This is similar to ReLU SAEs, except the sparsity loss is calculated as:

$$L_{\text{sparse}}(\boldsymbol{x}, s) = \lambda_s ||\boldsymbol{f}(\boldsymbol{x})||_{p_s}^{p_s} = \lambda_s \sum_i f_i(\boldsymbol{x})^{p_s} \tag{14}$$

**TopK SAEs** (Gao et al., 2025; Makhzani & Frey, 2014) enforce sparsity by retaining only the top $k$ activations per sample. The encoder is defined as:

$$\boldsymbol{f}(\boldsymbol{x}) := \text{TopK}(\mathbf{W}^{\text{enc}}\mathbf{x} + \mathbf{b}^{\text{enc}}) \tag{15}$$

where TopK zeroes out all but the $k$ largest activations in each sample. This approach eliminates the need for an explicit sparsity penalty but imposes a rigid constraint on the number of active latents per sample. An auxiliary loss $\mathcal{L}_{aux} = ||e - \hat{e}||^2$ is used to avoid dead latents, where $\hat{e} = W^{dec}z$ is the reconstruction using only the top-$k_{aux}$ dead latents (usually 512), this loss is scaled by a small coefficient $\alpha$ (usually 1/32).

**JumpReLU SAEs** (Rajamanoharan et al., 2024) replace the standard ReLU activation function with the JumpReLU activation, defined as

$$\text{JumpReLU}_\theta(z) := zH(z - \theta) \tag{16}$$

where $H$ is the Heaviside step function, and $\theta$ is a learned parameter for each SAE latent, below which the activation is set to zero. JumpReLU SAEs are trained using a loss function that combines L2 reconstruction error with an L0 sparsity penalty, using straight-through estimators to train despite the discontinuous activation function. A major drawback of the sparsity penalty used in JumpReLU SAEs compared to (Batch)TopK SAEs is that it is not possible to set an explicit sparsity and targeting a specific sparsity involves costly hyperparameter tuning. While evaluating JumpReLU SAEs, Rajamanoharan et al. (2024) chose the SAEs from their sweep that were closest to the desired sparsity level, but this resulted in SAEs with significantly different sparsity levels being directly compared. JumpReLU SAEs use no auxiliary loss function.

**BatchTopK SAEs** (Bussmann et al., 2024) impose a TopK constraint over the activations of entire batches during training. I.e. for a desired sparsity $k$ and a batch size $B$, all activations not within the top $B \cdot k$ in a batch are zeroed. During training, a threshold value is learned, which replaces the BatchTopK activation at inference time to avoid dependencies between test inputs.

## A.3. ITDA

### A.3.1. LATENT EXAMPLES

This section provides further detail of cherry-picked latents in a Llama 70B ITDA. The example prompts are randomly sampled within four ranges: $[2, \inf)$, $[0, 2)$, $[-2, 0)$, $(-\inf, 0)$. Prompts that do not activate latents are excluded from both the example prompt tables and the histograms. Activations collected over 1.28 million tokens from Gao et al. (2020).

**Latent 50**: Responds specifically to the token "How". The input string from which this activation was taken, with relevant token highlighted, is "How Idris Elba's 'Luther' Puts Us in the Mind set of a Renegade Detective. "Luther " is a series about righteous indignation . Yes, it's a police"

| Activation | Prompt |
|---|---|
| 4.331 | Q:How to handle puppeteer exception on synchronous execution |
| 4.331 | Q:How to change the trigonometric identity to sec |
| 4.331 | Q:How to write a React Stateless Functional Component in Typescript |
| 3.211 | in the US, Canada, UK or AustraliaHow do we promote your campaign?Giving Tuesday campaigns will |
| 2.879 | Remove User search limit per month PDO::MYSQLHow can I have PHP remove a users limit every month |
| 1.996 | line doesn't work. This Q & A How make autoscroll in Scintilla? has the |
| 1.973 | much is an Xbox One at Rent a Center? How much is a PS4?How much is an |
| 1.150 | redirect to https://www.test1.com content How can I konfigure it properly? Rewrites in |
| 1.042 | in Jesus. What difference would that make? And how would I know if I was really forgiven? A |
| 0.767 | an excellent example of this. In it she demonstrates how the concept of national identity fractures society, creating a |
| -0.742 | Dental Plans Association is a not-for-profit organization with some for-profit affiliates. We offer a nationwide package of |
| -0.874 | source. Our people. Your success. We understand the needs & ways to handle your backend SEO process and |
| -1.140 | As Kate Sheppard reported last week, July 26th marked |
| -1.647 | can get them for $180 dollars. Please come by my desk and give me your travel arrangements. |
| -2.272 | ? I saw the API. RallyRestApi restApi = new RallyRestApi(username, password, |
| -2.335 | Check out our new site Makeup Addiction No native support of previous PlayStation games |
| -2.468 | Most of the people at this detention centre in Tripoli will |
| -2.900 | Plus Free Shipping on Orders Over $50 Mario Badescu: Free Standard Shipping on $50+ |
| -3.615 | I'm working through a detox/cleansing phase and it's |

*Table 3.* Example prompts and activations for Latent 50

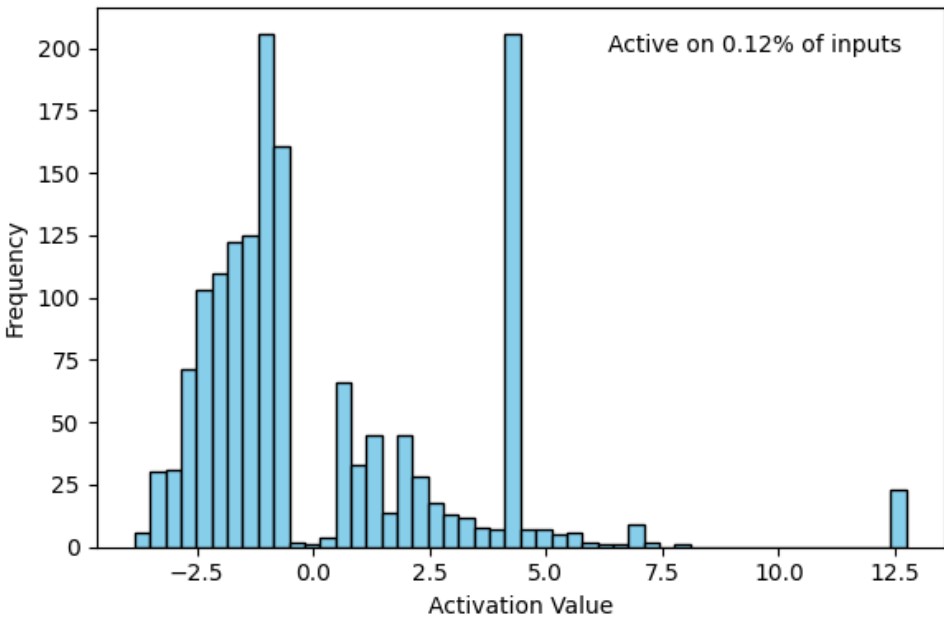

*Figure 3.* Activation distribution for latent 50. The peak at 4.33 corresponds to prompts starting with "Q: How". Activations of 0 are omitted for legibility.

**Latent 16990**: Responds tokens relating to surprise. The input string from which this activation was taken, with relevant token highlighted, is "web . You truly realize how to bring a problem to light and make it crucial . Many more people should really have a look at this and have an understanding of this side of the story . It 's surprising you 're not more prevalent , as you most really possess the gift . It is much easier to deal with the status quo than to push forward beyond the fear . He forwarded the email to her and"

| Activation | Prompt |
|---:|---|
| 5.482 | It astounds me, without exception to see how goats can climb. |
| 3.930 | in a host of colors, patterns 3 Surprising Reasons: Why The Large Floor Tiles Are So Popular |
| 3.775 | ulkan.html======gulpahum It's great that Google Android will support Vulkan. Now, the |
| 2.172 | Q: Unbelievable strange file creation time problem I have a very |
| 2.142 | you most really possess the gift. It is much easier to deal with the status quo than to push |
| 0.967 | could easily be confusing cause and correlation. Seems entirely possible |
| 0.923 | ) any later version. * This program is distributed in the hope that it will be useful, * but |
| 0.812 | 's movie we follow the beautiful model in a shockingly surreal journey through the rural countryside of Italy on a |
| 0.519 | /xhtml1/DTD/xhtml1-transitional.dtd"¿ = "http://www.w3.org/1999/xhtml |
| 0.000 | to, sometime soon. Even if it makes the difference between living and dying, there's just no way |
| -0.713 | So this guy was sucking brezz, felt something in his mouth, tasted somehow |
| -0.807 | expiratory manoeuvres in children: do they meet ATS and ERS criteria for spirometry |
| -0.851 | line, Mirage, and I put it on and felt my woman warrior emerge. Yeah, I know a |
| -0.864 | a strategical bombing. "We, as many others, we sign up in them in the Air Force |
| -1.081 | You are here Cigarette sales dive, hurting health |

*Table 4.* Activation values and corresponding prompts

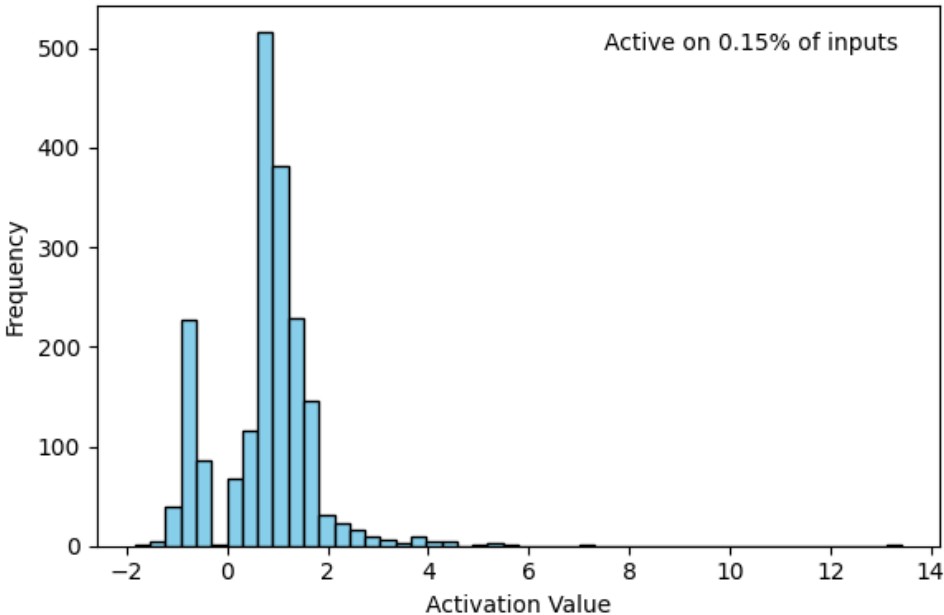

*Figure 4.* Activation distribution for latent 16990

**Latent 17002**: Responds tokens relating to surprise. The input string from which this activation was taken, with relevant token highlighted, is "the deadlines is our priority and we often acquire strict actions to fulfil our guarantee. We have significant knowledge in the sphere of homework online help; that's why we think We all know what sort of help a student requirements. Just invest in College assignments online and enjoy. Project Online is a flexible online solution for project portfolio administration (PPM) and each day get the job done. It enables corporations to get"

| Activation | Prompt |
| --- | --- |
| 2.899 | have significant knowledge in the sphere of homework online help; that's why we think We all know what sort |
| 2.447 | to fulfil our guarantee. We have significant knowledge in the sphere of homework online help; that's why we |
| 2.348 | at AOT you will receive the assistance and support that will help make your study time convenient, manageable and |
| 2.134 | to help your business succeed. We realize that where a company is located has a significant impact on its ability |
| 2.067 | This contains libraries that a) do not change frequently or b) require c |
| 1.128 | development acceleration and counter-propagating waves synchronization. A simple formula for SR pulse delay time evaluation is presented |
| 1.056 | the millions of dollars in moderately affected person venture funding that helps most nascent companies. All employers topic to |
| 0.926 | in Japan in 1954. Toru Kumon, a high school math teacher, was trying to |
| 0.900 | so we understand customers and we know what it's like to have full responsibility. We understand very clearly |
| 0.707 | in this connection, it is important that they use their time well. Trusted and Experienced Tutors |
| -0.000 | Mossy fiber sprouting after recurrent seizures during early |
| -0.714 | the study's authors. As the oldest and most commonly used method to screen for prostate cancer, rectal |
| -0.760 | to upgrade to new versions, I've been pretty much forced to jump to new manufacturers each time. I |
| -0.822 | Ugh! As i move things around to pack them or get rid of them, i am finding |

*Table 5.* Activation values and corresponding prompts for latent 17002

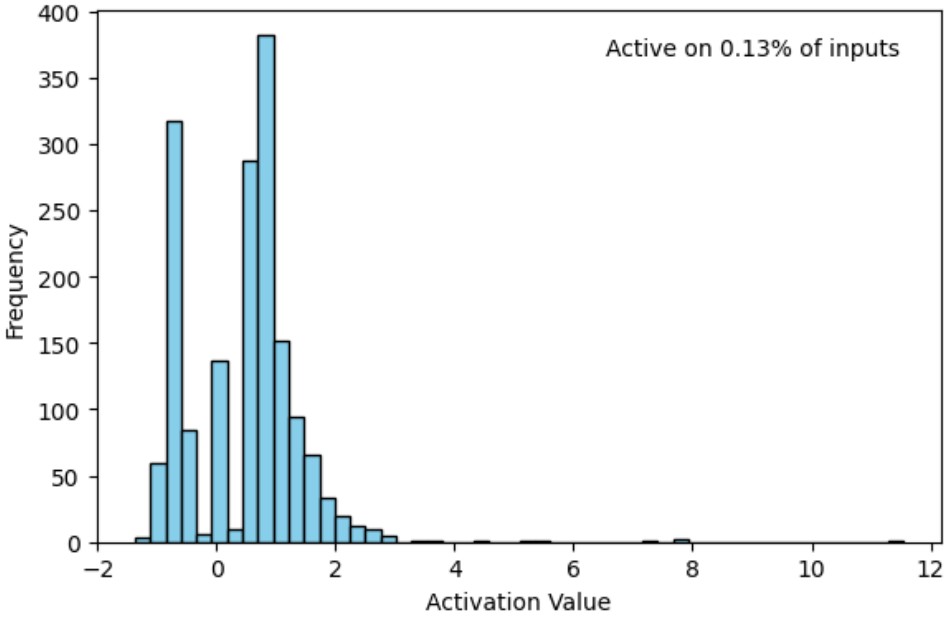

*Figure 5.* Activation distribution for latent 17002

A.3.2. Decomposition Examples

This section includes examples of decompositions of inputs in the Pile subset dataset, decomposed by an ITDA trained on layer 40 of Llama 3.1 70B with an L0 of 40. These prompts are taken from the test split, hence Sequences above 7000. Token 30 is chosen as it includes contextual information whilst keeping the prompt relatively short and readable. Activations less than 0.0001 are omitted. The text displayed in the third column is the prompt corresponding to the atom's activation with the specific token highlighted.

**Sequence 7006 Token 30:** "1 . Field of the Invention \n The present invention relates to a camera system for transmitting and receiving data to and from a camera by obtaining information /hlon"

| Atom Index | Activation | Atom Prompt |
|---:|---:|---|
| 8313 | 4.2174 | electronic imaging, and more particularly, to the use ==of== electronic imaging for processing financial documents, such as checks |
| 7878 | 2.7632 | member, in which a conductive portion is formed ==in== an insulator, the composite member being used in |
| 14123 | 2.5308 | a touch screen terminal of the other party successively ==scans== the multiple frequencies included in the proximity detection sequence. |
| 7864 | 2.1850 | by reference. The present invention relates to a method ==of== forming a composite member, in which a conductive |
| 5800 | 2.0224 | my pregnancy, I tried to gather as much information ==on== how painful labor might actually be. I would often |
| 4698 | 1.5757 | the wearer and thus to better conform to the body ==of== the wearer. Such extension and expansion about the wearer |
| 14124 | 1.3699 | the multiple frequencies included in the proximity detection sequence. ==If== signal strength at each frequency is greater than a preset |
| 5004 | 1.1926 | Gait properties change with age because of a decrease ==in== lower limb strength and visual acuity or knee joint |
| 9268 | 1.1632 | o login for bem sucedido salvar o nome ==de== usuário em uma variável e utilizar -l á em |
| 7889 | 1.0781 | insulator, the composite member being used in, ==for== example, a wiring board in the fields of electric |
| 6083 | 1.0443 | Learning Technology Programme (TLTP) from which ideas ==about== application and benefits came. The ideas from TLTP |
| 15748 | 1.0169 | Admin Properties an exception is raised, which list ==the== properties not supported. For more information see the |
| 6258 | 0.8658 | pumping, we examined the dependency of H+ pumping ==on== plant Pi status. Both H+ pumping and the |
| 15678 | 0.8496 | treewidth of the circuit as that of the graph ==representing== it; if we use associativity to rewrite |
| 9362 | 0.8415 | Retrieval ==of== blade implants with piezosurgery: two clinical cases |
| 12606 | 0.8150 | designed to provide users with voice communication services while they ==are== on the move. Current mobile communication systems are capable |
| 7903 | 0.7805 | wiring board in the fields of electric appliances, electronic ==appliances== and electric and electronic communication. The present invention also |
| 17194 | 0.7706 | -based visual programming tool. Node: The source definition ==for== nodes that can be used in Node-RED Fl |
| 5623 | 0.7612 | your recording is ready, you'll receive an update ==on== . . . your dashboard homepage with the playback link and the |
| 6929 | 0.7047 | def stateTimeThreadStart(): database.getTable ==('CLIENTS')== threads = [] threads.append(threading |
| 7231 | 0.6891 | + a poll I won't make claims as ==to== their gifts and charms, but H&M do |
| 7930 | 0.6539 | an insulating material that can be suitably used ==in== the manufacturing method of |
| 14704 | 0.6371 | The position and phase of the Moon are based ==on== the predictable motions of the Moon |

*Table 6.* Atom activations and corresponding text snippets for sequence 7006, token 30.

**Sequence 7008 Token 30:** Psych Studies is a website owned and maintained by Dr Andrew G . Thomas , Swansea University , UK . The purpose of the site is to host online ==question==

| Atom Index | Activation | Atom Prompt |
|---:|---:|---|
| 1208 | 3.9800 | ¡—begin_of_text—¿ Questions or Need Help Related to The Hunting Report Newsletter. Call |
| 4006 | 2.0558 | exponentially since its adoption over three decades ago. Recent questions have been raised regarding the cost-effect |
| 9803 | 1.5613 | xmlns :x si =' http :// www .w 3 .org / 200 1 /XMLSchema -instance ' xmlns =' http :// poly |
| 3301 | 1.3823 | : Using M - Test to show you can differentiate term by term. I have the series $\sum$ |
| 7704 | 1.2946 | top financial institutions have recommended in a new report that whistle -blowing be rewarded in an environment of growing |
| 1246 | 1.2604 | ¡—begin_of_text—¿ Question No: 51 You are developing a test |
| 8073 | 1.1217 | acid diet in rats. Within 3 h of ingesting an imbalanced amino acid diet (IAAD) |
| 8660 | 1.0420 | : sql queries and inserts I have a random question. If I were to do a sql select and |
| 8351 | 1.0366 | processing huge amounts of documents efficiently. Predictions that document payment methods would decline have not been realized. In |
| 6059 | 0.9382 | (CAL) and the importance of applying it in nurse education. The articles recognize the general technological developments as |
| 5258 | 0.8906 | from 15 to 99. Our ranks are made up of men and women; students and retirees; |
| 2480 | 0.8181 | play "Survival of the Tastiest" on Android, and on the web. Playing on the |
| 1 | 0.8178 | ¡—begin_of_text—¿ Q: Why was Mundungus banned from the Hog |
| 10958 | 0.7221 | lemek üzere tek zarfta iki pusulayla oyunu hem yurt içinde hem de yurt dışında kull |
| 2880 | 0.6843 | , which can be used to drag and drop 3D objects and characters into scenes. Amazon ... continue |
| 2345 | 0.6257 | ¡—begin_of_text—¿ Fractional isolation and chemical structure of hemicellulos |
| 9553 | 0.6178 | bunch of horrible experiences. It's about time I chronicled the peaks of my journey so far and this |
| 8865 | 0.5841 | er noen av bildene politiet fant på mobil telefonene til dem som ble pågrepet i |
| 5071 | 0.5709 | 76.1 (5.7) years ). Walking speed, cadence, stance time, swing |
| 3722 | 0.5641 | million fans then voted on the players using paper and online ballots. The top two vote-getters from each |
| 4928 | 0.5300 | and across primate taxa. As the problems of habituation become more obvious, the application of such indirect |
| 6133 | 0.5173 | to consider the benefits and costs of introducing computer programs as part of the teaching provision for nurses and other health |
| 3426 | 0.4858 | ette Sawyer Cohen, PhD, clinical assistant professor of psychology in pediatrics at Weill Cornell Medical College in |

*Table 7.* Atom activations and corresponding text snippets for sequence 7008, token 30.

**Sequence 7110 Token 30:** "Robot-assisted laparoscopic renal artery aneurysm repair with selective arterial clamping. Renal artery aneurysms represent a rare clinical"

| Atom Index | Activation | Atom Prompt |
|---:|---:|---|
| 9371 | 5.0731 | val of blade implants with piezosurgery: two clinical cases. In this work an ultrasound device was used |
| 9013 | 3.7738 | 7 ]]. Lack of response to IFX is a stable |
| 153 | 3.0721 | ¡—begin_of_text—¿ Clinical comparison of high-resolution with high-sensitivity collimators |
| 11632 | 2.2699 | dysesthesia. A new and relatively frequent side effect in antineoplastic treatment ]. Pal |
| 8988 | 1.9684 | 40% of patients who respond initially and achieve clinical remission inevitably lose response over time |
| 3944 | 1.7801 | Coronary artery disease (CAD) accounts for the largest number of these deaths. While efforts aimed at treating |
| 4391 | 1.6158 | believed to play a fundamental role in orthopedic research because bone itself has a structural hierarchy at the first |
| 10181 | 1.5146 | problems in women, the frequency with which primary care providers may encounter mental health problems, and issues of mental |
| 5322 | 1.2979 | return variable is always None So I found a strange thing that happens in Python whenever I try to return |
| 3981 | 1.2433 | have been directed toward prevention and rehabilitation. CAD is commonly treated using percutaneous coronary intervention (PCI), |
| 4509 | 1.1507 | 2:26 Prostate exams are potentially life-saving. But the process of getting one can be nerve |
| 4457 | 1.0142 | bone regeneration will be discussed. This unique 3 D tube-shaped nanostructure created by electrochemical anod |
| 14867 | 0.9638 | data or conclusions published herein. All content published within Cureus is intended only for educational, research, and reference |
| 6740 | 0.8502 | Adventure Time was already slated to be a LEGO Ideas official set, so it |
| 10227 | 0.8491 | (ADPLD), as traditionally defined, results in PLD with minimal renal cysts. Classically |
| 14150 | 0.8193 | of. It was (and is) a hearty baked meal—one to satisfy the hunger of very hard-working |
| 7014 | 0.7914 | a shelter for men with alcohol, drug, and mental health problems at 149 W. 132nd St |
| 3993 | 0.7646 | percutaneous coronary intervention (PCI), and this treatment has increased exponentially since its adoption over three decades ago |
| 8340 | 0.7237 | . Today's financial services industry is facing the immense challenge of processing huge amounts of documents efficiently. Predictions |
| 8500 | 0.6808 | taken a first step in understanding how to manipulate specific neural circuits using thoughts and imagery. The technique, which |
| 14836 | 0.6754 | ago, when frequencies below 100 Hz were considered extreme lows, and reproduction below 50 Hz was about |
| 15928 | 0.6624 | ¡—begin_of_text—¿ Late complications of radiotherapy for nasopharyngeal carcinoma. To evaluate and |
| 3910 | 0.6456 | Rationale and trial design. Cardiovascular disease (CVD) currently claims nearly one million lives yearly |
| 14869 | 0.6120 | by Pryce in 1946 in the Journal of Pathology and Bacteriology |

*Table 8.* Atom activations and corresponding text snippets for sequence 7110, token 30.

### A.3.3. MATCHING PURSUIT

The Matching Pursuit algorithms due to Mallat & Zhang (1993) is described in Algorithm 2, with the algorithm for iteratively constructing the dictionary of activations in Algorithm 1.

---

**Algorithm 2** Matching Pursuit (MP) with Normalized Dictionary

---

**Input:** Normalized dictionary $\mathbf{D} \in \mathbb{R}^{m \times n}$, batch of signals $\mathbf{X} \in \mathbb{R}^{B \times n}$, number of nonzero coefficients $L$
**Output:** Coefficient matrix $\mathbf{C} \in \mathbb{R}^{B \times m}$
Initialize the residuals: $\mathbf{R} \leftarrow \mathbf{X}$
Initialize the coefficients: $\mathbf{C} \leftarrow \mathbf{0}_{B \times m}$
**for** $\ell = 1$ to $L$ **do**
 Compute correlations: $\mathbf{Corr} \leftarrow \mathbf{R}\,\mathbf{D}^{\mathsf{T}}$
 For each $b \in \{1, \ldots, B\}$, find the best atom index: $j_b \leftarrow \arg\max_j \big|\mathbf{Corr}_{(b,j)}\big|$
 Let $c_b \leftarrow \mathbf{Corr}_{(b,j_b)}$ (the max absolute correlation for sample $b$)
 Update coefficients: $\mathbf{C}_{(b,j_b)} \leftarrow \mathbf{C}_{(b,j_b)} + c_b$
 Update residuals: $\mathbf{R}_{(b,:)} \leftarrow \mathbf{R}_{(b,:)} - c_b\,\mathbf{D}_{(j_b,:)}$
**end for**
**return** $\mathbf{C}$

---

### A.3.4. SAE INTERPRETABILITY METRICS

In this section we present results from SAEBench (Karvonen et al., 2024a) for three LLMs: the 70m and 160m parameter variants of Pythia (Biderman et al., 2023), and the 2b parameter variant of Gemma 2 (Team et al., 2024). Our experiments were conducted with the 0.4.0 beta release of SAEBench. The SAEs with which we compare with are a mix of architectures: generally, the worst performing SAE is a ReLU SAE (Bricken et al., 2023; Cunningham et al., 2023), whilst the best is a top-k SAE (Gao et al., 2020) or p-annealing SAE (Karvonen et al., 2024b).

SAEBench metrics were created for evaluating SAEs, which limits their applicability to ITDAs. In particular, Spurious Correlation Removal and Targeted Probe Perturbation make assumptions about the decoder weight matrix of SAEs that do not apply to ITDAs and so we do not include those results. We include sparse probing results as they are, but note that applying the top-k operation to ITDA latent activations is not comparable to applying it to SAE activations due to negative activations and normalization of the decoder. SAEs furthermore have weak performance at identifying human-interpretable concepts on probing tasks, when compared to other methods (Kantamneni et al., 2025). Automated interpretability approaches for SAEs transfer directly to ITDAs, but their usefulness in measuring interpretability is questionable (Heap et al., 2025). Given the challenges benchmarking SAEs, and the additional difficulty of applying those metrics to ITDAs, we advise strong caution in interpreting these results and look to advances in SAE benchmarking to better evaluate the interpretability of ITDAs. For now, we emphasize the evidence provided by the representation similarity results in Section 5.

**Sparse probing** assesses whether SAEs capture specific predefined concepts. For each concept (e.g., sentiment), the k most relevant latents are identified by comparing their average activations on positive versus negative examples. A linear probe is then trained on these top k latents. If the latents align closely with the concept, the probe achieves high accuracy, even though the SAE was not directly trained to represent that concept. We evaluate on $k \in 1, 2, 5, \infty$ latents to handle cases where concepts are distributed over multiple latents.

In **automated interpretability** for each chosen latent, a language model generates a "feature description" based on a variety of activating examples. During testing, a dataset is assembled by sampling sequences that trigger the latent at varying levels of activation, along with randomly selected control sequences. The LLM then uses its generated description to predict which sequences are likely to activate the latent, and the accuracy of these predictions defines the interpretability score.

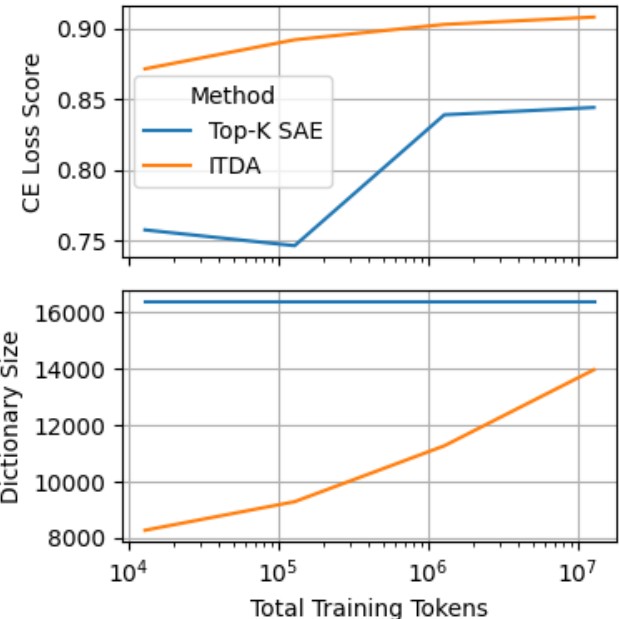

*Figure 6.* Performance of Pythia-70m SAEs and ITDAs when trained on limited numbers of tokens. Here, we do not crop the size of the dictionary, as on smaller numbers of training tokens the dictionary does not achieve the minimum size.

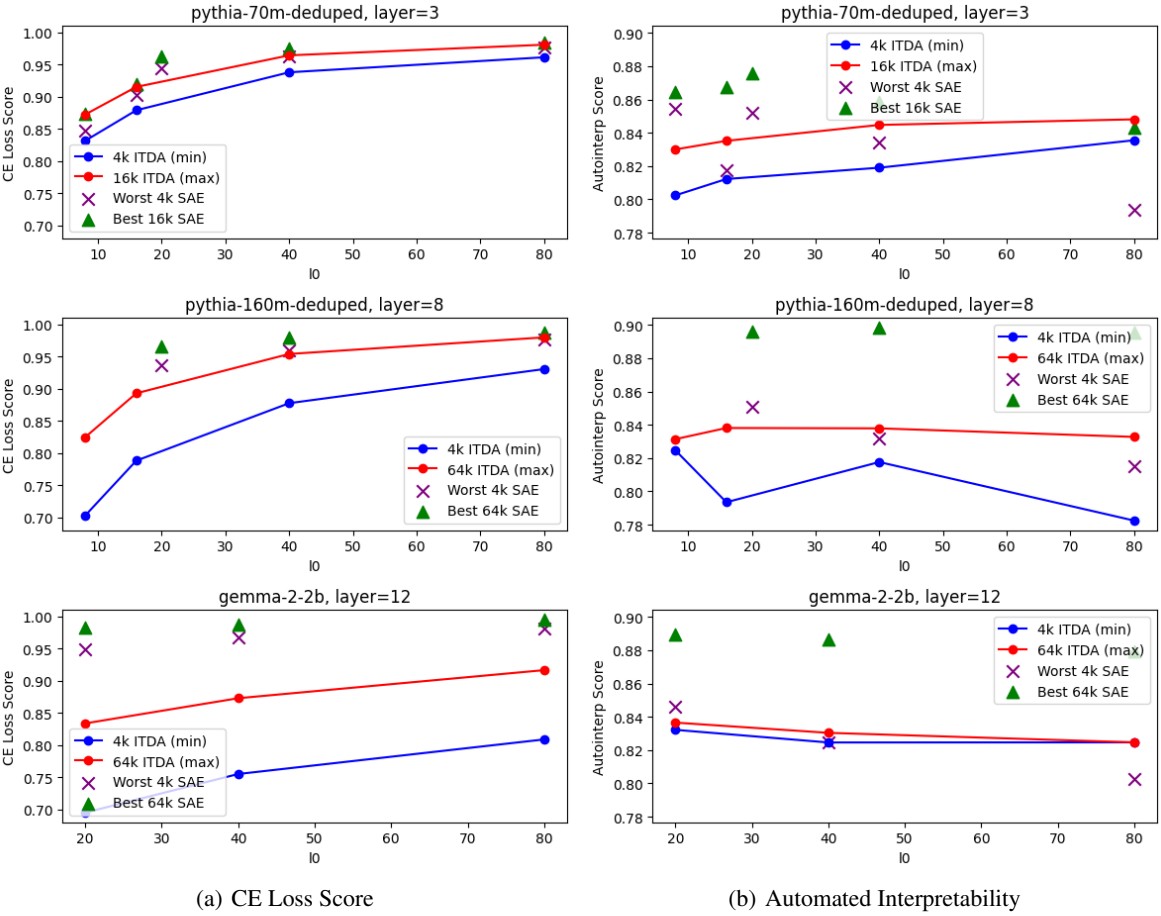

(a) CE Loss Score            (b) Automated Interpretability

*Figure 7.* CE loss score and automated interpretability scores. CE loss score is defined in **??**, see Karvonen et al. (2024b) for details of the automated interpretability metric.

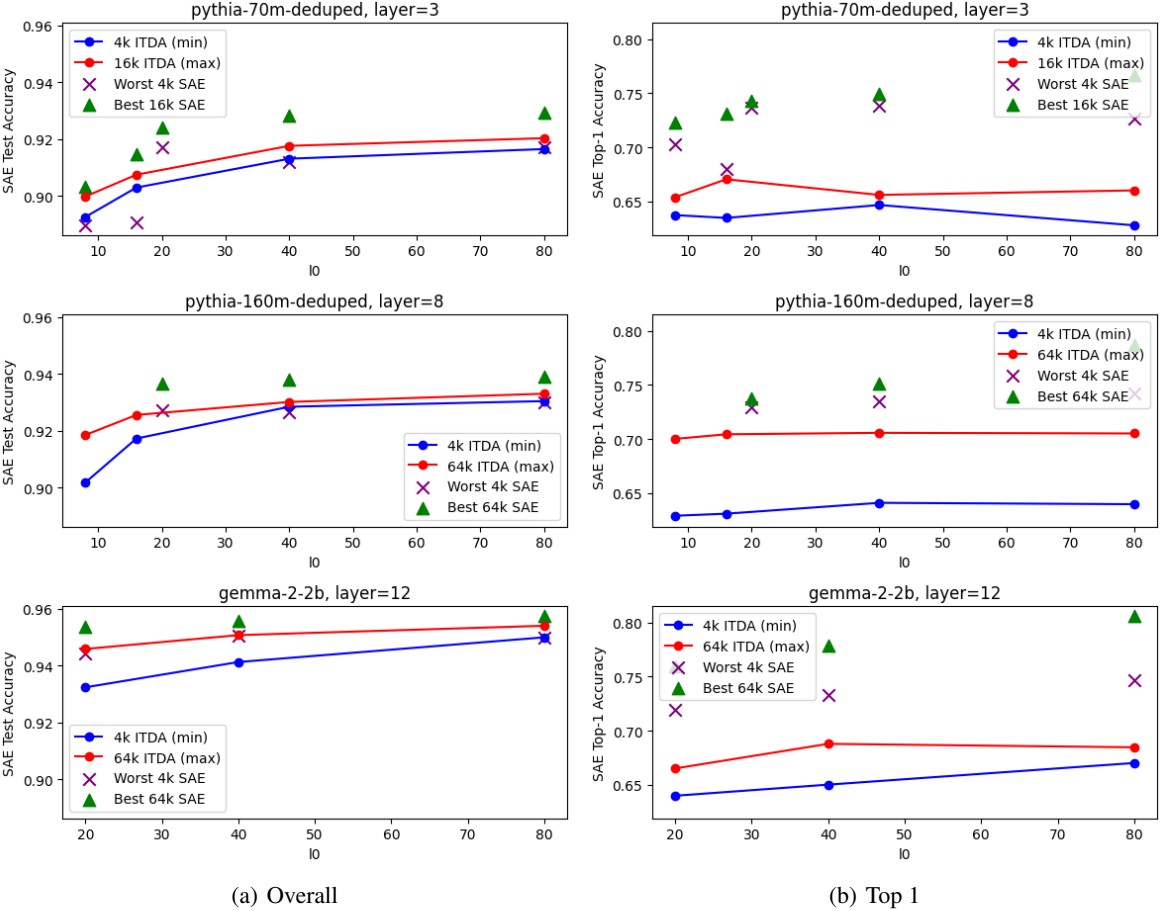

(a) Overall

(b) Top 1

*Figure 8.* Overall and top-1 linear probing scores. Overall score measures whether concepts can be probed from the activations of all the latents, whilst top-1 score measures whether the concepts can be probed from the single most active latent.

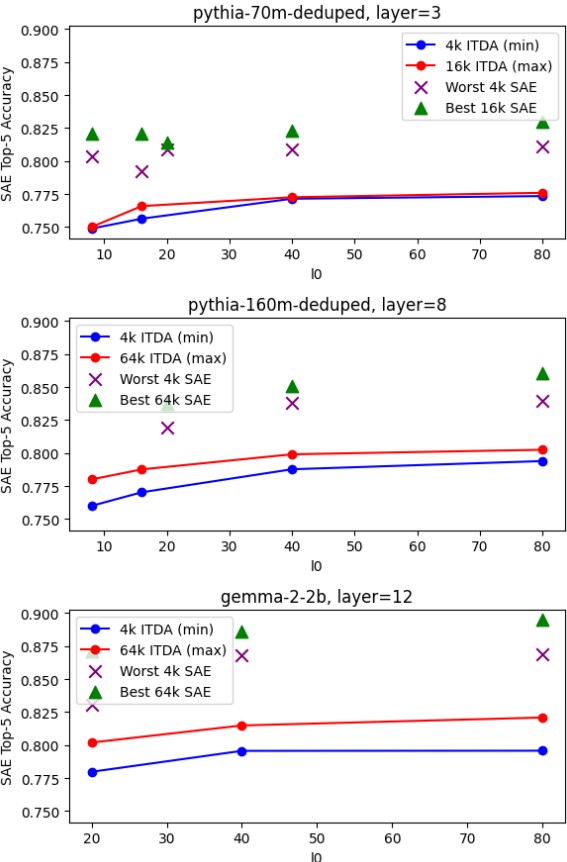

*Figure 9.* Top 5 sparse probing scores.

## A.4. Representation Similarity

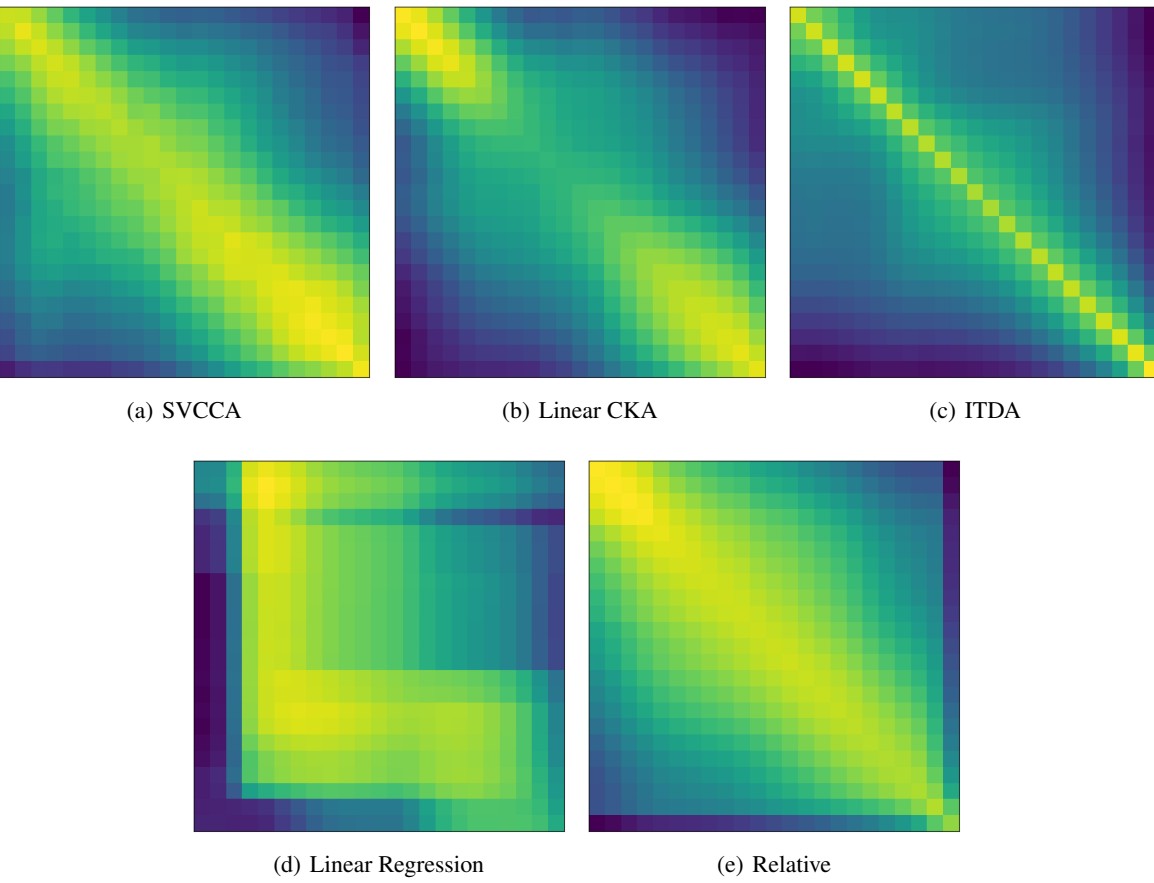

(a) SVCCA          (b) Linear CKA          (c) ITDA

(d) Linear Regression          (e) Relative

*Figure 10.* Heatmap of similarity indexes between layers in instances of GPT-2 medium with random initialisations, metric average across all pairs of model. See Figure 5 for GPT-2 small results and further explanation of the plots.

### A.4.1. MODEL SIMILARITY

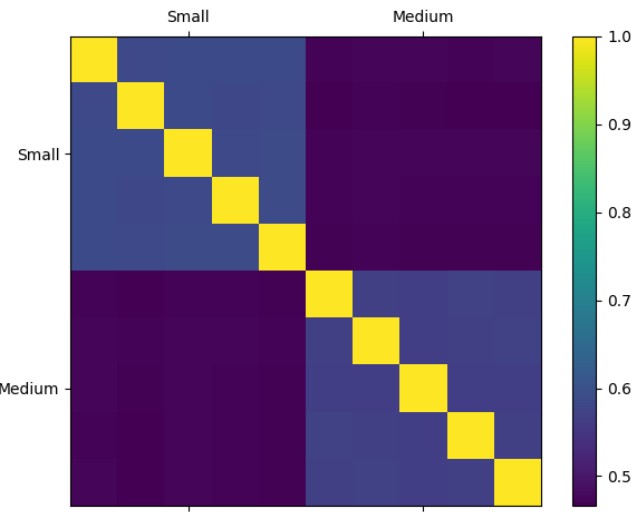

*Figure 11.* Model similarity between ITDAs trained on instances of GPT-2 small and medium. Note the higher similarity within model architectures.

### A.4.2. LAYER CONVERGENCE DURING TRAINING

Raghu et al. (2017) demonstrate using SVCCA that, in a convnet and resnet trained on CIFAR-10 (Krizhevsky et al., 2009), the representations of early layers converge earlier during training than the later layers. Martinez et al. (2024) replicate this result on the Pythia suite of language models (Biderman et al., 2023), using the CKA metric from Kornblith et al. (2019).

For models in the 70m, 160m, and 410m Pythia models, we trained ITDAs on each non-terminal layer every ten thousand training steps, and calculated the ITDA representation similarity from Equation 8 for each checkpoint with respect to the last checkpoint, which are presented in Figure A.4.2. The similarity metrics for the first two layers converge to 1 during the last third of training, meaning that the ITDA dictionary is stable across those checkpoints. No other layers converge, with the penultimate similarity measure decreasing for successive layers.

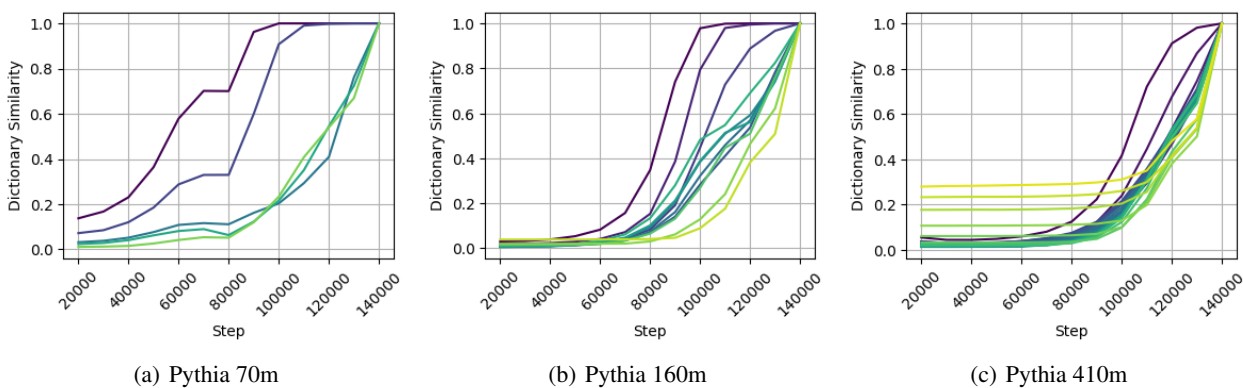

| (a) Pythia 70m | (b) Pythia 160m | (c) Pythia 410m |

*Figure 12.* ITDA similarity between layers every 10k steps during training compared to their final state. Layers increase from the first layer in dark blue to the last layer in light yellow. Note the convergence of early layers to 1 before the end of training.

