# OpenReview forum: "Inference-Time Decomposition of Activations (ITDA): A Scalable Approach to Interpreting Large Language Models"
_ICML.cc/2025/Conference — ICML 2025 poster_

### Official Review · Reviewer_5pSy · 2025-03-03

**Overall Recommendation:** 4

**Summary:**

The paper introduced a novel method for decomposing language model activations into interpretable features that could replace SAE with matching pursuit. Thanks to its efficient and fast-converging matching pursuit algorithm (and not having to perform expensive training of NNs), this enables scalable learning about the features of an extremely large language model. While it's 10% short of the reconstruction performance of the SAE, it still suggests scalable, good approach that can be also transferred to other models.

## update after rebuttal
Although this paper provides a highly efficient proof-of-concept alternative to the current SAE, I believe there is room for improvement that validates the actual usefulness of the proposed method. The main concern is given the limitation and skeptics of the validity of current SAEs, the proposed method is a weak, faster version of them, which raises even more doubts about the validity. Providing more concrete evidence of the validity and usefulness of the proposed method will make the paper stronger. Still, I have decided to increase the score to "Accept", given the potential of the method.

**Claims And Evidence:**

The paper claims ITDA is a faster, scalable alternative to SAEs for interpreting LLMs. It says ITDAs train 100-1000x faster, needing just 0.1-1% of the data—like 1 million tokens versus 16 billion for SAEs—and can handle huge models like Llama-3.1 70B and 405B in minutes on a single GPU. This checks out with examples like training 55 ITDAs on GPT-2 in under 2 hours versus 250 hours for SAEs. The efficiency and scalability feel solid, though exact times and hardware details would help.

**Essential References Not Discussed:**

All relevant papers were cited

**Experimental Designs Or Analyses:**

They train ITDAs on Pythia-70m’s residual stream, comparing cross-entropy loss scores to SAE variants like ReLU and TopK. The formula—measuring loss degradation against zero-ablation—is clear, and plots show ITDAs plateau at 1M tokens while SAEs improve with more. They also test ITDA’s Jaccard similarity against SVCCA and CKA, matching layers across model instances. Sample size (five instances) is reasonable, and results favor ITDA. And then they track ITDA similarity during training on Pythia 70m-410m, showing early layers stabilize first. No issues here.

**Methods And Evaluation Criteria:**

ITDA’s method builds a dictionary of activations greedily, using Matching Pursuit for sparse coding, an alternative to traditional SAE’s gradient descent. It makes sense for fast, interpretable decomposition, especially for big LLMs. They evaluate with cross-entropy loss for reconstruction, automated interpretability and sparse probing for features, and Jaccard similarity for model comparisons, all tested on datasets like the Pile and models from Pythia to Llama. Using established benchmarks like SAEBench keeps it grounded. My only gripe is the interpretability metrics feel vague; tighter criteria there would sharpen the story.

Recently, there is a paper "Sparse Autoencoders Can Interpret Randomly Initialized Transformers" that questions the validity of autointerp, so maybe it's not too robust to use it to meausre the performance of ITDA. This is the fundamental problem shared by the literature (not only this paper), however.

**Other Comments Or Suggestions:**

N/A

**Other Strengths And Weaknesses:**

The paper took a bold leap to replace SAE with a classical matching pursuit algorithm. This is very novel and its benefits (reduced training cost with a slight inference overhead and reasonable reconstruction loss) are impressive. Considering that this paper is one of the first papers to try this approach, this paper reasonably provides all the necessary, important insights already. As mentioned by the paper, ITDA maybe cannot fully replace the SAE due to its worse reconstruction loss, but it could be useful for way larger models by providing a rough insights into the model. However, at the same time, this raises a question on how useful ITDA actually is. Also, it seems the examples in the appendices show that the latent often activates for semantically different tokens sometimes. (Sequence 7008 Token 30. Nurse vs Question vs Fraction, etc.)

**Questions For Authors:**

Q1. How would this technique be used for activation steering? Would there be any difference to using SAE?

Q2. What would be the effect of initializing the dictionary from a different dataset? There is a paper "Sparse Autoencoders Trained on the Same Data Learn Different Features" that shows that SAE is very sensitive to a random seed, which affects the weight initialization. Similarly, would initializing the dictionary with different activations lead to finding different features?

Q3. What does "[the dictionary] is associated with a prompt and token index, rather than being learned from the activations of a specific model"? It shows up repeatedly across the paper, but it is not clear.

Q4. "Sparse Autoencoders Can Interpret Randomly Initialized Transformers" shows that random and trained transformers produce similarly interpretable SAE latents. Would ITDA simiarly lead to this weird outcome or could it be more robust?

**Relation To Broader Scientific Literature:**

The paper introduced a very big leap from the gradient-based SAE to a classical matching pursuit algorithm. Less reliance on learning the activations of the model using gradients helps the method to better generalize to other models. It would be interesting to extend it to more applications that are being studied in the literature like activation steering, safety, etc. as a future work to further see its validity and utility.

**Theoretical Claims:**

N/A

---

> ### Author Rebuttal · Authors · 2025-04-01
>
> Thank you for your feedback and we appreciate that you feel this work is novel and “its benefits are impressive”.
>
> > The efficiency and scalability feel solid, though exact times and hardware details would help.
>
> We ran our experiments on a range of hardware so it is hard to provide precise summary statistics about resource usage. We will release a W&B project alongside the camera-ready paper that will include granular information about hardware and run-times.
>
> > My only gripe is the interpretability metrics feel vague; tighter criteria there would sharpen the story.
>
> Thanks for this. We have expanded the exposition and explanation of interpretability results, and include a description of each of the metrics that we hope makes it easier for the reader to quickly understand the results. However, these benchmarks are quite complex so we feel it is better to point the reader to the SAE Bench paper than try to provide a detailed explanation here. The explanations are similar to those in the SAE Bench paper, but we don’t have space to reproduce them here.
>
> > Recently, there is a paper "Sparse Autoencoders Can Interpret Randomly Initialized Transformers"... [also Q4]
>
> Thanks for highlighting this, SAE benchmarks are a new field and rapidly evolving so you’re right to question the validity of the results. The paper you reference found that SAEs trained on randomly initialised transformers learn single token features that are deemed to be interpretable. The same token in different contexts will likely have similar representations even in a random model, so it is not surprising that the SAE learns token features. These single token features are then clearly interpretable, as they only respond to a very specific context (presence of the token).
>
> This isn’t the case with ITDAs, however. We draw the reviewer's attention to the analysis of Latent 17002 in the Appendix, which is an example of multi token feature that activates on contexts relating to homework in contrast to the single token latents in that paper. On a subsample on 10k inputs taken from the Pile, we calculated on how many unique tokens each latent is active and plotted it here: https://imgur.com/mm3l6jE.
>
> > latent often activates for semantically different tokens sometimes…
>
> For these decompositions please bear in mind that the anchor text is only one part of interpreting the behaviour of the latent. While eg. "Nurse" may be the token activation that is used in the decomposition, it's possible that the contribution due to this token is a broader medical concept (which aligns with the context of the prompt) rather than a concept relating specifically to nurses.
>
> > Q1. How would this technique be used for activation steering? Would there be any difference to using SAE?
>
> Yes, it would be used in the same way as SAEs, i.e. modifying the latent activations at inference time. We’re unsure of the usefulness of SAE steering, however, so haven’t included any results relating to this in our paper.
>
> > Q2. What would be the effect of initializing the dictionary from a different dataset? …
>
> This is a good point, and one that we haven’t explored in our paper: we use The Pile throughout our experiments. ITDA dictionaries greedily select as we iterate over the dataset, so shuffling the dataset will likely lead to a substantially different decoder. We suspect data ordering will have some effect on the reconstruction and interpretability performance of ITDAs, and we would hope to explore this in follow-up work.
>
> > Q3. What does "[the dictionary] is associated with a prompt and token index, rather than being learned from the activations of a specific model"? It shows up repeatedly across the paper, but it is not clear.
>
> We agree that this wasn’t clear in the original paper, so we’ve added a system-level diagram (https://imgur.com/a/UIfzcdX) and some explainer text to clarify: “The elements in the dictionary can be viewed from two perspectives. For the purpose of decomposition using matching pursuit, they are absolute activation vectors taken from the model; this perspective is used when describing the algorithms for constructing ITDA dictionaries and for decomposing activations. Alternatively, the dictionary can be viewed as a collection of prompts and references to tokens in those prompts, in combination with part or all of a model. For example, a prompt may be ``The cat sat on the carpet.", the token reference is to the second token ``cat", and the partial model is the first 5 layers of GPT-2. The absolute activation of this element is the activation for the second token in the prompt after the 5th layer in GPT-2. We use this perspective when comparing dictionaries between models, as the prompt and token reference are model-agnostic.”
>
> > (Q4 addressed above)
>
> Thank you again for your thoughtful and constructive review—we hope that our clarifications and revisions have addressed your concerns, and would be grateful if you might consider updating your score accordingly.

---

> > ### Comment · Reviewer_5pSy · 2025-04-07
> >
> > Thank you for the detailed response. I appreciate the clarifications and additional visualizations, especially the distinction between single-token and multi-token features. While the expanded explanations help, my core concern about the validity of the interpretability remains. I maintain my current score of Weak Accept.

---

### Official Review · Reviewer_miuH · 2025-03-10

**Overall Recommendation:** 3

**Summary:**

The paper introduces Inference-Time Decomposition of Activations (ITDA) as a fast and scalable alternative to SAEs for interpreting LLM activations. ITDA constructs a dictionary of representative activations using matching pursuit, allowing it to be trained 100-1000× faster than SAEs with only 0.1-1% of the data, making it feasible for models as large as Llama 3.1 405B. Despite its efficiency, ITDA achieves 90% of the reconstruction performance of SAEs and performs comparably on interpretability benchmarks such as sparse probing and automated interpretability. Unlike SAEs, ITDA enables direct cross-model comparisons, leading to a new representation similarity metric that outperforms SVCCA and CKA in layer-matching tasks. The paper demonstrates ITDA’s potential for analyzing large-scale LLMs, tracking layer convergence, and identifying model behavioral shifts, making it a promising tool for mechanistic interpretability at scale.

**Claims And Evidence:**

The paper presents strong empirical evidence for most of its claims, particularly in demonstrating the efficiency and scalability of ITDA compared to SAEs.

**Essential References Not Discussed:**

n/a

**Experimental Designs Or Analyses:**

The experimental design is generally sound, with appropriate comparisons between ITDA and SAEs using SAEBench, sparse probing, and automated interpretability. The cross-entropy loss evaluation effectively measures reconstruction performance, and the layer-matching task provides a reasonable test for representation similarity. However, the comparison to SAEs is somewhat limited, as it primarily focuses on ReLU SAEs, while more advanced variants (e.g., TopK, P-Annealing SAEs) are only briefly discussed.

**Methods And Evaluation Criteria:**

The proposed method, ITDA, is well-motivated as a scalable alternative to SAEs for LLM interpretability. The use of matching pursuit for sparse coding is reasonable, and the greedy dictionary construction aligns with the goal of efficient feature extraction. Evaluation is conducted on standard benchmarks, including SAEBench, sparse probing, and automated interpretability, which are appropriate for assessing interpretability performance. The representation similarity evaluation using layer-matching tasks follows established methods (SVCCA, CKA) and provides meaningful comparisons.

**Other Comments Or Suggestions:**

Langage -> language in the abstract line 13

**Other Strengths And Weaknesses:**

Strengths:
1. ITDA is 100-1000x faster than SAEs, with only 0.1-1% data required.
2. ITDA enables representation similarity analysis, outperform SVCCA and CKA.

Weakness:
1. No quantitative interpretability results for larger models.

**Questions For Authors:**

1. The paper mentions that a lower loss threshold leads to better reconstruction but a larger dictionary. How does this affect interpretability performance?
2. The paper mentions negative activations but does not explore their meaning. Do they correspond to specific model behaviours?

**Relation To Broader Scientific Literature:**

The paper extends prior work on mechanistic interpretability and sparse dictionary learning, particularly addressing the computational limitations of SAEs. ITDA builds on classical dictionary learning methods by using matching pursuit for inference-time optimization, enabling efficient decomposition of LLM activations. It also introduces an ITDA-based Jaccard similarity measure, improving upon existing representation similarity metrics like SVCCA and CKA in layer-matching tasks. Additionally, ITDA aligns with recent efforts in model diffing by enabling cross-model comparisons without requiring gradient-based training. While it offers scalability advantages, further empirical comparisons with advanced SAEs would strengthen its positioning in the literature.

**Theoretical Claims:**

The paper does not contain formal proofs but relies on algorithmic descriptions and empirical validation. The theoretical foundation of ITDA, particularly its use of matching pursuit for sparse coding and greedy dictionary construction, aligns with established methods in dictionary learning.

---

> ### Author Rebuttal · Authors · 2025-04-01
>
> Thank you for your comments, we appreciate your recognition of the strong evidence for the “efficiency and scalability of ITDA in comparison to SAEs”.
>
> > However, the comparison to SAEs is somewhat limited, as it primarily focuses on ReLU SAEs, while more advanced variants (e.g., TopK, P-Annealing SAEs) are only briefly discussed.
>
> Thanks for raising this. To clarify, we did evaluate those variants as extensively as ReLU SAEs: In our experiments we compare against all SAE variants that have been evaluated by SAEBench. This includes ReLU, TopK, P-Annealing, Gated, and JumpReLU SAEs. ITDAs almost always have worse reconstruction performance than the best performing SAE variant. However, different SAE variants learn different kinds of features [1], and have different interpretability properties, so the SAE with the best reconstruction may not always be the best choice. As such, in our plots we show the SAE with the overall best reconstruction performance, and the best performing ReLU SAE. This provides a baseline for reconstruction performance without strawmanning SAEs by selecting badly optimised instances.
>
> [1] Hindupur, Sai Sumedh R., et al. "Projecting Assumptions: The Duality Between Sparse Autoencoders and Concept Geometry." arXiv preprint arXiv:2503.01822 (2025).
> [2] https://www.saebench.xyz/
>
> > No quantitative interpretability results for larger models.
>
> One of the major advantages of ITDAs is that we were able to train them on large models like Llama 70B and 405B on widely available hardware, unlike SAEs which require considerably more computational resources. However, this means that SAEBench does not support (or currently need to support) multiple GPUs, which would be necessary for Llama 405B. Furthermore, there would be no publicly available SAEs with which to compare the results. We could run experiments on Gemma 2 9B, as there are open source SAEs for this model, if you think this would considerably strengthen the paper (but this would probably not be possible during the discussion phase).
>
> > The paper mentions that a lower loss threshold leads to better reconstruction but a larger dictionary. How does this affect interpretability performance?
>
> Increasing the dictionary size from 4k to 16k on Pythia models, and from 4k to 64k on Gemma 2 2B, results in a modest improvement on autointerp and sparse probing benchmarks. We have now included an appendix section going into considerably more detail on these interpretability benchmarks, for a range of dictionary sizes and L0s.
>
> > The paper mentions negative activations but does not explore their meaning. Do they correspond to specific model behaviours?
>
> This is a good question and one that we’ve not extensively investigated. Zero-ablating negative latent activations impacts reconstruction performance, but we don’t have a good understanding of how negative latent activations affect model behaviour. If we interpret strong positive activations as meaning an input is strongly related to a feature, then strong negative activations could mean an input is strongly unrelated to a feature, which is hard to validate. We emphasize the rarity of strong negative activations, however. Our appendix latent examples were cherry-picked to include strong negative activations, but they are rare (<0.001%) in general. SAE latents frequently have small positive activations, so we are not particularly worried about the small positive or negative activations in ITDAs.
>
> We hope this addresses your questions and we are keen to improve your faith in, and support of, this paper. Please let us know if you have any further questions or concerns. If these clarifications have addressed your concerns, we would be grateful if you might consider revisiting your overall score.

---

> > ### Comment · Reviewer_miuH · 2025-04-03
> >
> > Thank you for providing responses to my concerns. After reading your response, I decide to keep my score.

---

### Official Review · Reviewer_w5VN · 2025-03-12

**Overall Recommendation:** 2

**Summary:**

The main idea of this paper is to apply a dictionary learning approach to the problem of finding sparse representations of activation.  ITDA builds the dictionary at inference time. The algorithm works by first trying to reconstruct the activation from the atoms in the dictionary. Reconstruction is done by Matching Pursuit (thus scales at least linearly in dictionary size). If the reconstruction is possible with a sufficiently small number of atoms (a hyperparameter) the latent representation is accepted, if not, the $\mathbf{x}$ is added to the dictionary. In this way, as we repeatedly iterating and updating the dictionary during inference time.

**Claims And Evidence:**

The authors provide some experiments of their approach. In Section 3 they report some loss as a function of $\ell_0$ norm, which makes sense and is consistent with my expectations.

They also provide SAE benchmarks, and report some positive scores in the text, but I am not aware of the validity of these benchmarks.

In Section 4 they also consider representation similarity, which is an interesting new topic. I didn't fully understand the claims they were making in this section, or what the implications are.

**Essential References Not Discussed:**

.

**Experimental Designs Or Analyses:**

See methods and Evaluations.

**Methods And Evaluation Criteria:**

I think there needs to be more explanation of the experiments. Specifically, Section 4 is a bit confusing to me. This might be because of other misunderstandings I have.

**Other Comments Or Suggestions:**

There is interchanging use of terms like "atom", "token" and "activation". The authors need to be more precise here.  This paper could also benefit from a system-level diagram?

You should define what $\mathbf{D} \cup \\{ x\\}$ means.

Can you be a bit more rigorous when defining *everything*. For example, what is the CE Loss with respect to? You don't need to go over basic definitions of course, but you need to provide enough information that I can quickly figure out what you are doing.

**Other Strengths And Weaknesses:**

Overall the idea of applying iterative dictionary learning tools for learning SAEs is a good idea, however, I find that the paper is inconsistent in its language, and I think this work would benefit from more time spent on the presentation and evaluation.

**Questions For Authors:**

What is the difference between $x$ and $\mathbf{x}$?

**Relation To Broader Scientific Literature:**

ITDA is, as far as I can tell, a pretty unique way to learn an SAE, and some approach like this might be faster that traditional approaches.

**Theoretical Claims:**

No theoretical claims are being made.

---

> ### Author Rebuttal · Authors · 2025-04-01
>
> Thanks for your feedback and suggestions, we’re glad you like the idea and are keen to improve the presentation.
>
> > In Section 4 they also consider representation similarity, which is an interesting new topic. I didn't fully understand the claims they were making in this section, or what the implications are.
>
> We claim that the Jaccard Similarity between two ITDA dictionaries is a simple, performant, and state-of-the-art measure of representation similarity and outperforms existing methods on a layer matching task on GPT-2 model variants. We propose that this approach opens up exciting research directions in finding differences between models, as we can take the difference between ITDAs as well as their intersection.
>
> For two models with pre-trained ITDAs with dictionary D0 and D1 sizes n and m, our method allows for measuring similarity in O(n + m) time. In comparison, SVCCA and CKA require learning a map between representation spaces, and the relative representation measure due to [1] requires comparing computing a dataset of activations and comparing to a set of anchor points. Furthermore, that the intersection of D0 and D1 accurately tracks model similarity suggests that the difference between D0 and D1 accurately tracks differences between models. This opens exciting research directions in “model diffing”, where the goal is to find differences between models, for example before and after fine-tuning.
>
> > I think there needs to be more explanation of the experiments. Specifically, Section 4 is a bit confusing to me.
>
> Thanks for this; we have added more explanation of experiments, particularly in section 4, and we feel this has strengthened the paper and made it more readable for a general audience. Here is a summary of the changes as the full updated section is too long to reproduce here:
> Further explanation of the significance of the results and comparison to additional methods
> Greater explanation of the layer matching task, including a formula for the calculation of the scores to remove all ambiguity
> Moved the layer convergence experiments to the appendix as they did not directly evidence the value of ITDA here
> Added a model-level matching task to emphasize the usefulness of ITDAs for differentiating between models.
>
> > There is interchanging use of terms like "atom", "token" and "activation". The authors need to be more precise here. This paper could also benefit from a system-level diagram?
>
> Thanks for raising this. One of the primary advantages to ITDAs is that concepts like “atom”, “token”, and “activation” are actually interchangeable: ITDA dictionaries atoms are activations, but they’re also token references. We appreciate that this can be confusing though, so we’ve added a system-level diagram (https://imgur.com/a/UIfzcdX) and some explainer text to clarify: “The elements in the dictionary can be viewed from two perspectives. For the purpose of decomposition using matching pursuit, they are absolute activation vectors taken from the model; this perspective is used when describing the algorithms for constructing ITDA dictionaries and for decomposing activations. Alternatively, the dictionary can be viewed as a collection of prompts and references to tokens in those prompts, in combination with part or all of a model. For example, a prompt may be "The cat sat on the carpet.", the token reference is to the second token "cat", and the partial model is the first 5 layers of GPT-2. The absolute activation of this element is the activation for the second token in the prompt after the 5th layer in GPT-2. We use this perspective when comparing dictionaries between models, as the prompt and token reference are model-agnostic.”
>
> Our work draws on research from the fields of sparse dictionary learning, mechanistic interpretability, and representation similarity. Each of these fields uses its own terminology, and we’ve stuck to those conventions. For example, it doesn’t make sense to refer to activations when discussing representation literature. We’ve added a glossary to the appendix to help the reader (but don’t have enough characters remaining to reproduce it here).
>
> > Can you be a bit more rigorous when defining everything.
>
> Yes, absolutely. In particular, we have updated the methodology in section 3, including the CE score formula and “D ∪ {x}”,  and improved the explanation of the experiments in section 4.
>
> > What is the difference between…
>
> These are the same and this was a formatting mistake - we have corrected this, thanks for pointing this out.
>
> We again thank the reviewer for their helpful comments, and hope our changes are satisfying. Regrettably the character limit of this response means that we cannot reproduce the changes to the paper here, but we are happy to do so during the discussion period. If these changes have been satisfying, we politely ask the reviewer to reconsider their score.
>
> [1] Moschella, Luca. "Latent communication in artificial neural networks." ICLR (2024).

---

> > ### Comment · Reviewer_w5VN · 2025-04-02
> >
> > Some of the information included in this rebuttal goes a long way towards understanding the ITDA. The authors certainly should  include something like what they included in rebuttal (excerpt below) in the main paper:
> >
> > > “The elements in the dictionary ... reference are model-agnostic.”
> >
> > I see other reviewers also had some confusion related to this e.g. Q3 of 5pSy
> >
> > I disagree with your decision to use "atom", "token" and "activation" interchangeably depending on the context when discussing elements of the dictionary. It is confusing, and in my opinion, mathematically incorrect. They are **not** the same thing.
> >
> > **Acceptable options for presentation**:
> >
> > 1.  **Atoms are activations**. Each activation happens to correspond to a token from a prompt, but these are not atoms themselves. You could **call the prompts/tokens "interpretable labels" of your atoms**, akin to SAE literature. Then you can explain how the set of labels can be used to construct new dictionaries for different LLMs as desired. This will fall more neatly into a linear dictionary learning framework.
> >
> > 2. If you **insist on referring to promts/tokens as atoms**, you should **define your (nonlinear) forward model** appropriately. Something like:
> >
> > $$\hat{\mathbf{x}} = \sum_{i} a_i f_{\ell}(\mathbf{d_i}),$$ where $f_{\ell}$ is an is the "partial LLM" that you mentioned in the system diagram.  In practice, $f_{\ell}(\mathbf{d_i})$ would be precomputed, so everything is still linear.
> >
> > With these clarifications in mind, I did a second read of this paper, and I feel that any misunderstandings I had are primarily due to the way ideas are presented, and not lack of care on my part. For example, statements like this cause confusion:
> >
> > > ITDA dictionaries consist of prompts and token indices, rather than learned atoms,
> >
> > You also "learn" atoms, just in an online fashion. Secondly, as I mentioned above, I think this is a misuse of the term atom.
> >
> > *System Diagram*: This is a good start, but I think it would be good to also include how you can "transfer" dictionary from one LLM to the another using the label prompts.
> >
> > **I stand by my evaluation of this paper. After clarification from the authors, I am even more confident in my assessment that the presentation is poor. While it is possible that some of these issues could be addressed in the camera ready version, I believe there is too much work to do, and the submitted version falls below an acceptable level of rigor for ICML. A re-evaluation after significant revisions is warranted, thus I will maintain my score.**

---

> > > ### Author Response · Authors · 2025-04-03
> > >
> > > Thank you for your response. We appreciate that you found the additional information in the rebuttal helpful in understanding our method, as well as your earlier comments about the technical merit of the paper. We think your first option for improving the presentation of the paper is good, and we will include those changes in future versions of the paper.

---

### Official Review · Reviewer_XACq · 2025-03-13

**Overall Recommendation:** 4

**Summary:**

The paper proposes a new algorithm for mechanistic interpretability as an alternative to sparse autoencoders. Their algorithm iteratively identifies new token activations to add as dictionary items based on their similarity to the current dictionary. If the similarity is too low (i.e. reconstruction through the dictionary is not accurate enough), the (contextualized) token is added as a new item to the dictionary. Hence, the dictionary items are effectively identifiable from token indices into a prompt. This allows the construction of a new representation similarity measure based on the Jaccard index computed over two dictionaries.
While the reconstruction performance of the proposed model is worse, it performs on par with ReLU-SAEs on interpretability benchmarks, but worse than the state-of-the-art Top-K SAEs. However, the proposed method is significantly cheaper to train than SAEs, making it applicable to large open source models with hundreds of billions of parameters. Finally, on a model layer similarity measurement task, the proposed method outperforms previous methods.

**Claims And Evidence:**

The support for the claims made is generally convincing. There is also support for the claim that the proposed method is a lot more efficient than SAEs. However, I would like to see a more thorough investigation of the relation between training data size and other hyperparameters of the proposed method with the training time, in comparison to SAEs.

**Essential References Not Discussed:**

The discussion of related literature in this paper is substantial.

**Experimental Designs Or Analyses:**

I checked the soundness of the experimental designs and they appear solid.

**Methods And Evaluation Criteria:**

The evaluation of interpretability seems to rely on a recent benchmark and compares to state-of-the-art models. The baselines for the model instance layer similarity task seem quite old (2017 & 2019), but I am unsure if a more recent one (Lan et al., 2024) is applicable in this case. The authors should clarify this.

**Other Comments Or Suggestions:**

line 389: "an efficient alternative"

**Other Strengths And Weaknesses:**

The paper uses only 7 out of 8 available pages and thus there is significant room for improvement. For example, some of the graphs from the appendix could be moved to the main body.
I would also like to see an investigation of how the choice of prompts etc. influences the interpretability and representation similarity results.

**Questions For Authors:**

None

**Relation To Broader Scientific Literature:**

The key contributions are related to the mechanistic interpretability literature, specifically the training of sparse autoencoders (SAEs) for large language models. SAEs are extremely expensive to train for large models and therefore investigations are often limited to small models such as GPT-2. The proposed method is reported to be several orders of magnitude faster to train while achieving similar results to some earlier SAEs. Given that the proposed architecture differs significantly from SAEs, it is plausible that there is a lot of room for improvement from future work. Hence, this work opens up meaningful new research directions.

**Theoretical Claims:**

There are no theoretical claims.

---

> ### Author Rebuttal · Authors · 2025-04-01
>
> Thank you for your positive and thoughtful feedback. In particular we appreciate your recognition of the evidence for the claim that this approach is “a lot more efficient than SAEs” and that you feel this work “opens up meaningful new research directions”.
>
> > However, I would like to see a more thorough investigation of the relation between training data size and other hyperparameters of the proposed method with the training time, in comparison to SAEs.
>
> We agree this is an important avenue for investigation. In our experiments, the ITDAs were trained on 1.28 million tokens, while the SAEs used for benchmarking ranged from 500 million to 16 billion tokens. This offers insight into the performance of ITDAs on low data settings across target models. A comprehensive exploration of this relationship, however, would require training new SAEs, a process that would take weeks for models like Gemma 2, which unfortunately exceeds our timeframe for revisions. Nevertheless, we will clearly highlight this limitation in our revised manuscript and suggest it as a critical area for future research.
>
> > The baselines for the model instance layer similarity task seem quite old (2017 & 2019), but I am unsure if a more recent one (Lan et al., 2024) is applicable in this case. The authors should clarify this.
>
> This is true, while CKA and SVCCA are standard approaches, they are not recent. Lan et al. proposes applying CKA to SAE decoder matrices as a measure of representation similarity, however applying this approach would require training 170 SAEs on the LLMs which would take 1-2 months of GPU time. Instead, we have added a more recent relative representation measure due to Moschella, Luca. "Latent communication in artificial neural networks." ICLR (2024). This measure also significantly outperforms SVCCA and CKA, but not our ITDA IoU method.
>
> Metric GPT-2 Small GPT-2 Medium
>
> Linear Regression (baseline) 0.16 0.07
>
> SVCCA (Raghu et al., 2017) 0.50 0.44
>
> Linear CKA (Kornblith et al., 2019) 0.69 0.61
>
> Relative (Moschella et al., 2022) 0.87 0.78
>
> ITDA (ours) 0.88 0.89
>
> Note that for better reproducibility we have replaced our self-trained Pythia instances with two sets of public GPT-2 model instances of different sizes.
>
> > However, since there is significant space left (1 page), it could also be moved to the main body to make the paper more self-contained.
>
> Following feedback from reviewers, we have expanded the main body with a system-level diagram and more explanation of the method and experiments. Consequently, there is now no space to move the examples and algorithm to the main body of the paper. We chose to prioritise these clarifications to help improve the reader’s understanding of the core methodology.  If there are specific supplementary plots or analyses you believe would significantly strengthen the main body, please let us know—we will explore adjustments accordingly.
>
> > I would also like to see an investigation of how the choice of prompts etc. influences the interpretability and representation similarity results.
>
> This would be an interesting direction for further exploration. SAEs are highly susceptible to shifts in the distribution of their training data, so it seems likely this is also the case for ITDAs. The interpretability pipeline is time-consuming to run, so we won’t be able to get these results during the review period. However, we think these experiments would meaningfully strengthen the paper and will add them to a future version.
>
> > line 389: "an efficient alternative"
>
> Good spot, thanks!
>
> We hope that this has addressed your concerns with the paper, and if so, would ask you to consider increasing your support of this paper.

---

> > ### Comment · Reviewer_XACq · 2025-04-07
> >
> > Thank you for your response. I will keep my favorable score.

---

### Decision · Program_Chairs · 2025-05-01

**Decision:**

Accept (poster)

**Comment:**

The paper tackles a significant practical problem in mechanistic interpretability: the prohibitive cost of training SAEs for large models. The ability to train dictionaries on models like Llama 405B in minutes on a single GPU represents a substantial practical advance and opens up possibilities for studying models previously inaccessible to typical academic resources. The efficiency gains come at the cost of reconstruction performance, consistently lagging behind SAEs. Given that ITDA is a demonstrably weaker approximation (in terms of reconstruction) of SAEs, and considering the ongoing debates and skepticism regarding the ultimate validity and meaningfulness of SAE-derived features themselves. Reviewers raised significant concerns about the clarity of the presentation, inconsistent terminology (atom/token/activation), and lack of rigor in definitions, while the authors made efforts to address this in the rebuttal.